# A cleavage rule for selection of increased-fidelity SpCas9 variants with high efficiency and no detectable off-targets

Péter István Kulcsár [1], András Tálas[1], Zoltán Ligeti [1,2,3], Eszter Tóth[1], Zsófia Rakvács[1], Zsuzsa Bartos [1], Sarah Laura Krausz[1,4,5], Ágnes Welker[6,7], Vanessza Laura Végi[1,4], Krisztina Huszár[1,7] & Ervin Welker [1,2] ✉

*Streptococcus pyogenes* Cas9 (SpCas9) has been employed as a genome engineering tool with a promising potential within therapeutics. However, its off-target effects present major safety concerns for applications requiring high specificity. Approaches developed to date to mitigate this effect, including any of the increased-fidelity (i.e., high-fidelity) SpCas9 variants, only provide efficient editing on a relatively small fraction of targets without detectable off-targets. Upon addressing this problem, we reveal a rather unexpected cleavability ranking of target sequences, and a cleavage rule that governs the on-target and off-target cleavage of increased-fidelity SpCas9 variants but not that of SpCas9-NG or xCas9. According to this rule, for each target, an optimal variant with matching fidelity must be identified for efficient cleavage without detectable off-target effects. Based on this insight, we develop here an extended set of variants, the CRISPRecise set, with increased fidelity spanning across a wide range, with differences in fidelity small enough to comprise an optimal variant for each target, regardless of its cleavability ranking. We demonstrate efficient editing with maximum specificity even on those targets that have not been possible in previous studies.

Although many challenges remain to be addressed until advances of the CRISPR technology can be translated into routine clinical practice, recent reports on both in vivo and ex vivo CRISPR-based gene therapy reaching the stage of clinical trials mark the enormous potential of CRISPR nucleases[1–10]. The *Streptococcus pyogenes* Cas9 (SpCas9) is the most frequently used nuclease for genome engineering with the highest potential for therapeutic applications amongst all RNA-guided nucleases of the type II CRISPR system. Tremendous research effort has been devoted to increase the potential of SpCas9 by minimizing its off-target activity, which poses safety concerns for its use in areas where high specificity is a requirement, e.g., in clinical applications[4,11–13]. Several methods have been developed to increase its

specificity including the application of double nickases[14,15] and dimer FokI fusion variants of SpCas9[16–18], single-guide RNAs (sgRNAs) with truncated or extended spacers[18–21], as well as mutant SpCas9 variants[4]. However, none of these have managed to fully eliminate off-target cleavage and/or preserve efficient on-target editing universally for most targets. One of the most promising approaches among these methods to decrease off-target activity has been the generation of increased-fidelity nuclease variants. A non-exhaustive list of these variants includes the rationally designed[21–25] (e.g. eSpCas9, SpCas9-HF1, HypaSpCas9 and Blackjack), as well as variants developed by a selection scheme[26–31] (e.g. evoSpCas9, Sniper, and HiFi). A number of variants have also been developed by combining mutations from

[1]Institute of Enzymology, Research Centre for Natural Sciences, Budapest, Hungary. [2]Institute of Biochemistry, Biological Research Centre, Szeged, Hungary. [3]Doctoral School of Multidisciplinary Medical Science, University of Szeged, Szeged, Hungary. [4]Biospiral-2006 Ltd, Szeged, Hungary. [5]School of Ph.D. Studies, Semmelweis University, Budapest, Hungary. [6]Institute of Cognitive Neuroscience and Psychology, Research Centre for Natural Sciences, Budapest, Hungary. [7]Gene Design Ltd, Szeged, Hungary. ✉e-mail: welker.ervin@ttk.hu

existing increased-fidelity nucleases (IFNs), these include e-plus, HF1-plus, HypaR, and HeFSpCas9[21,24,32,33]. We prefer to collectively refer to the variants as 'increased-fidelity' nucleases instead of 'high-fidelity' nucleases, because the term 'high-fidelity' have been reserved specifically for the SpCas9-HF variants[23], and also because, as it will be clear from this paper, they possess widely varying fidelities. While increased-fidelity variants greatly improve the potential for highly specific genome modifications, their limitations have also become increasingly apparent. Each of them initiate the editing of many targets with considerable off-target effects[21–28,30,31,33] and they exhibit increased target-selectivity, i.e., the variants do not initiate editing or they only do to a decreased extent at numerous target sites that are otherwise cleavable by the wild type (WT) SpCas9[21,23,24,31,33–35]. Our former in cellulo study also revealed that HeFSpCas9, one of the highest fidelity variants, cleaves a few targets only, albeit with high fidelity, however, these exact targets are the ones, that get cleaved by the eSpCas9 and SpCas9-HF1 variants with the most concomitant off-target effects[21,33]. This finding prompted us to investigate whether this pattern is also a characteristic of other increased-fidelity variants and target sequences.

In this study we demonstrate that (i) IFNs can be ordered according to their fidelity/target-selectivity, which has also been demonstrated using a smaller set of IFNs for a large number of on-target and off-target sequences in ref. 34. and ref. 31, respectively. Even more interestingly, we found that target sequences also fall into an order according to their cleavability by the variants. Our experiments suggest that target sequences have a distinct, albeit variable, activating effect on the editing process that is exerted in the same manner and at the same step of the SpCas9 cleavage mechanism as fidelity-increasing mutations and mismatches. Ultimately, mainly these sequence contributions control whether an IFN cleaves a target or not, and they also primarily determine the extent of their actual off-target propensity. For optimal, both highly efficient and specific editing, one should find an IFN with a fidelity/target selectivity ranking that is well matched to the sequence contribution of the target, i.e., the variant should have an activity that is sufficient to efficiently cleave the target sequence but insufficient to cleave any of its off-target sequences. (ii) The fidelity requirement of the potential target sequences is frequently not accounted for by the available variants. Therefore, to provide a near-optimal variant for any potential target, we generate additional variants to build the CRISPRecise set of IFNs with increasing fidelity with small enough differences between the variants to cover a wide range in high resolution. (iii) Using this knowledge and an extended set of variants, we project that practically every target can be edited without detectable genome-wide off-target effects (defined here as detectable by GUIDE-seq), by applying target-matched IFNs. We challenge this claim by testing, to the best of our knowledge, all known problematic target sites from the literature that have been unsuccessfully tried by the previously developed, commonly used SpCas9 IFNs[20,23,24,26,28], as off-target editing was still detected by GUIDE-seq.

## Results

### Cleavage rule controls the on-target activity of increased-fidelity nucleases

First, using an EGFP disruption assay in N2a cells, we compared the on-target activity of WT and seven IFNs; Blackjack-, e-plus, HF1-plus, Hypa-, HypaR-, evo- and HeFSpCas9 on 50 targets (target sequences can be found in Supplementary Data file 1) using flow cytometry (gating examples in Supplementary Fig. 1, results in Supplementary Fig. 2a–i)[21,24,26,32,33]. The results are also presented on a heatmap depicting disruption activities for each target, normalized to the wild type value in order to neutralize the effect of the cellular context and factors, such as sgRNA expression levels and sequence specificity of the NHEJ DNA repair system (Supplementary Fig. 2j). The variants exhibited varying normalized average on-target activity on these targets, Blackjack SpCas9 showing the highest, approaching that of the

wild type, and HeFSpCas9 showing the lowest. We found that the cleavage pattern is far from being random. By reordering variants based on the number of targets they cleave (Supplementary Fig. 2k), we noticed that, generally, when a target was cleaved by a nuclease, it was also cleaved by all other lower ranking nucleases (i.e., by all those variants that, in aggregate, cleave a larger number of targets in the set). Moreover, when we reordered the targets based on the number of variants that could cleave them (Supplementary Fig. 2l), we realized that, generally, when a nuclease cleaves a particular target, it also cleaves all other targets that are in higher position in the cleavability ranking. This particular pattern of results requires that the following three conditions are met: (i) There is a factor that determines the cleavability of the target sequences, and this factor is approximately a fixed value for each target sequence. (ii) There is a factor that determines the inhibitory effect of IFNs, which is a specific value for each IFN. (iii) The relationship between the magnitude of these two factors determines which IFNs will cleave the target and which will not. We named this phenomenon the *cleavage rule* of the targets and variants. In the particularly striking pattern the *cleavage rule* creates the cleaved and non-cleaved values are separated into two distinct classes in the two-dimensional cleavage map (Supplementary Fig. 3a). Binary classification confirms that the actual data of the two-dimensional cleavage map in Supplementary Fig. 3a tightly follow this cleavage rule (G-mean score of 0.987, for details see Methods section and Supplementary Data file 4) containing hardly any outliers. ROC curve (receiver operating characteristic curve) shows the fitting of the cleavage data of each variant to the two-dimensional cleavage map arranged according to the cleavage rule in Supplementary Fig. 3a, confirming that this rule applies for each variant (Supplementary Fig. 3b).

### On-target and off-target activities of IFNs on a given target are interconnected

To find out how this cleavability characteristic of the sequences is related to the off-target propensities of the variants, we conducted a mismatch screen. In this, we tested three PAM-distal positions each containing all three possible single mismatches tested as a mixture for each of the eighteen selected target sequences altogether with 162 mismatching sgRNAs (Supplementary Fig. 3c). In the case of all the IFNs, the specificity of the editing on a given target clearly depends on the position of the target within this ranking. The fidelity-increasing mutations in a given variant may reduce the activity of SpCas9 appropriately for a relatively small fraction of the target sequences, so that it cleaves the on-target sequences efficiently and exclusively, without cleaving the off-targets. This can be illustrated by the example of HypaSpCas9. Efficient cleavage with maximum specificity can be seen at targets 8, 15 and 34. However, target sequences from lower cleavability ranks, such as targets 7, 11 and 35, will not be cleaved at all and targets from higher cleavability ranks, such as targets 2, 3 and 5, are cleaved but with off-targets (Supplementary Fig. 3c).

Taken together, these results suggest that there are 3 main factors, namely target sequence contribution, mismatches and fidelity-increasing mutations, that collectively determine whether an IFN will cleave a target or any of its off-targets. Our results also imply that these three main factors affect SpCas9 activity in a similar way. Former studies have shown that both the mismatches and the fidelity increasing mutations in eSpCas9 and SpCas9-HF1 loosely trap SpCas9 in a catalytically inactive intermediate state and slow down the transition of the HNH domain to the active state[24,36–40]. Mismatches, both PAM-proximal and PAM-distal when bona fide off-targets are engaged, inhibit the HNH domain transition and tend to keep the nuclease in this inactive state[38–42]. Therefore, this is likely the step that is also affected by the contribution of the target sequence. This is supported by the facts that it is the formation of the hybrid helix between the spacer and the target DNA strand that activates the HNH domain transition[24,36,37,41,43], and that DNA cleavage efficiencies scale with the

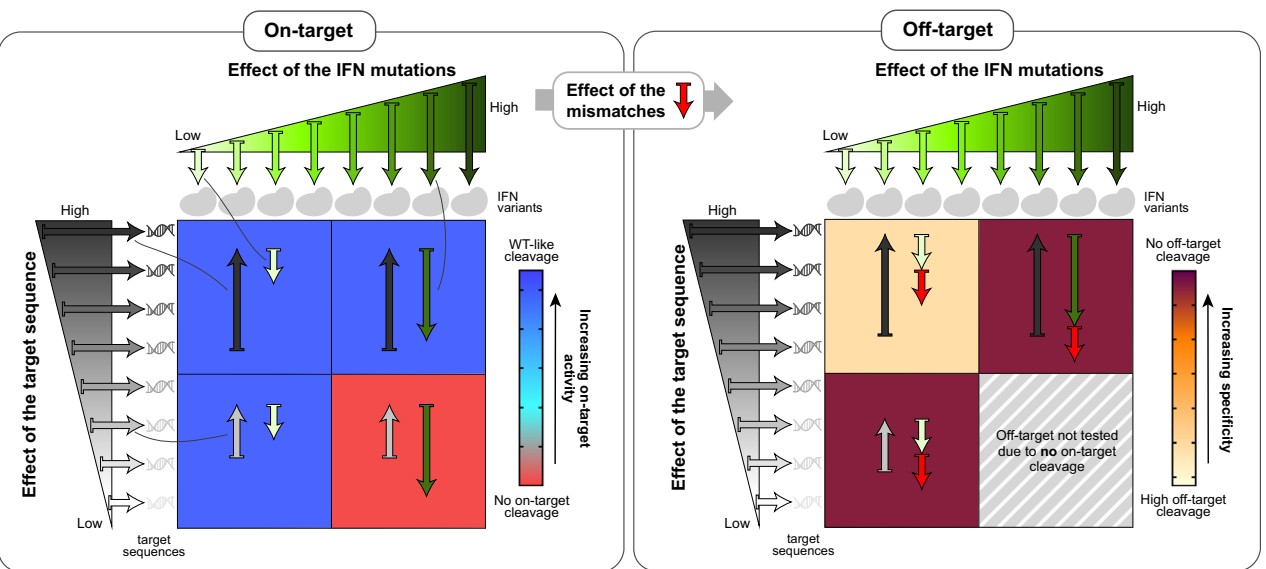

**Fig. 1 | Simplified explanatory figure of the 'cleavage rule'.** A simplified explanatory figure of the "cleavage rule" is presented to interpret the results shown in Fig. 2 and Supplementary Figs. 2, 3. Three main factors, namely target sequence contribution (effect of the target sequence), mismatches and fidelity-increasing mutations, that collectively determine whether an IFN will cleave a target or its off-targets. The coloring of the heatmap in this figure corresponds to that of the heatmaps in other figures within the manuscript. The left panel shows the on-target activity of two IFNs from different fidelity ranks on two targets from different cleavability ranks. When the activating effect of the target sequence contribution is larger than the inhibitory effect of the fidelity-increasing mutations, the SpCas9 variant cleaves the target (blue background), but when it is smaller, then the variant does not cleave the target (red background). The right panel shows the effect of a mismatch on the activity of the IFNs from the first panel on the same targets. When the activating effect of the target sequence contribution is larger than the combined inhibitory effect of the fidelity-increasing mutations and the mismatches, the SpCas9 variant cleaves the off-target sequence (yellow background), but when it is smaller, the variant does not cleave (burgundy background). In the case of the optimal, target-matched IFN the inhibitory effect of the fidelity-increasing mutations is only slightly smaller than the activating effect of the target sequence, so that it can still effectively cleave the on-target sequence, but when the effect of a mismatch is added, the combined inhibitory effect exceeds the contribution of the target sequence, and therefore it does not cleave any off-target. The effect of the same mismatch can vary in different sequential contexts[41], but for simplicity, here we apply the same effect in all of the example cases.

extent to which the HNH domain samples an activated conformation[44]. Thus, altogether, our results suggest that different target sequences can activate this transition to different extent resulting in the target ranking and the characteristic pattern on the heatmaps of Supplementary Fig. 2. Figure 1 shows how the sum of the effects of these three factors, i.e., the activating effect of the target sequence and the inhibiting effects of fidelity-increasing protein mutations and mismatches at the off-target sites, affect SpCas9 cleavage.

Our data also show that there are substantially larger differences in the effect of sequence contributions of different targets than the effect of some off-target mismatches, thus no single IFN is capable of the off-target-free cleavage of all sequences. The cleavage rule also imposes that in order to efficiently edit a target with the highest possible specificity, we need to select the IFN with the highest fidelity that still yields sufficient cleavage required for the given application. Supplementary Fig. 3d shows that applying this principle substantially increases the specificity of efficient IFN editing, however, several targets are still edited with considerable off-target effects (showing up to 20–70% of the on-target disruption values).

**Building a large set of IFNs with appropriate fidelities**
We hypothesize that maximal fidelity can be achieved universally for every target sequence by having an extended set of IFNs with increasing fidelity. These IFNs should cover a wide range of fidelity levels with sufficient resolution to provide an appropriate variant for targets from any cleavability rank. To test this idea, we made use of our prior discovery that Blackjack mutations in SpCas9 variants not only make the 5'G extension of sgRNAs more tolerable, but they also increase their fidelity to some extent[21]. By generating additional variants we established a set of 19 IFNs in total, including Sniper, HiFi, e-, -HF1, Hypa-, HypaR-, evo-, HeF-, their Blackjack counterparts (indicated

with a 'B' prefix), e-plus, HF1-plus and Blackjack SpCas9[21–23,27,28,32]. We found that all newly added variants fit in the pattern seen in Supplementary Fig. 3 when tested on the on-target and mismatch screens (Fig. 2). Containing hardly any outliers (G-mean score of 0.984) they all strictly follow the cleavage rule (Fig. 2a). When by taking advantage of the cleavage rule, the highest fidelity variant with sufficient activity is used for each target (Fig. 2e), the specificity of IFN-editing is substantially increased compared to the rest of the IFNs. In addition, overall, a higher specificity could be reached using this set than with the set of only 7 IFNs seen earlier in Supplementary Fig. 2d (Fig. 2e). The 20–70% normalized off-target edits seen in Supplementary Fig. 2d are effectively diminished by using this set of 19 variants suggesting that they approximate an appropriate resolution (Fig. 2e). The SpCas9 variant with the highest fidelity rank from our set of 19 IFNs that still show sufficient activity on a given target is hereafter referred to as target-matched variant for that given target. These results suggest that maximal fidelity can be reached universally by using an appropriate set of SpCas9 variants with small enough differences in fidelity that can provide an optimal target-matched IFN to every target from any position of the cleavability ranking.

**The cleavage rule appears to be universal**
To validate our findings, (i) we assessed the mismatch tolerance by genome-wide off-target detection instead of a mismatch screen, (ii) tested another cell line and used NGS instead of a disruption assay and (iii) analyzed data from a large target library, as described below.

(i) To validate that the characteristics of IFNs revealed by mismatch screening reliably reflect their genome-wide off-target effects, we performed GUIDE-seq analyses using various IFNs on 4 EGFP targets from different cleavability ranks of Fig. 2 in HEK293.EGFP cell line. Supplementary Fig. 4 (and Supplementary Fig. 5) shows that the

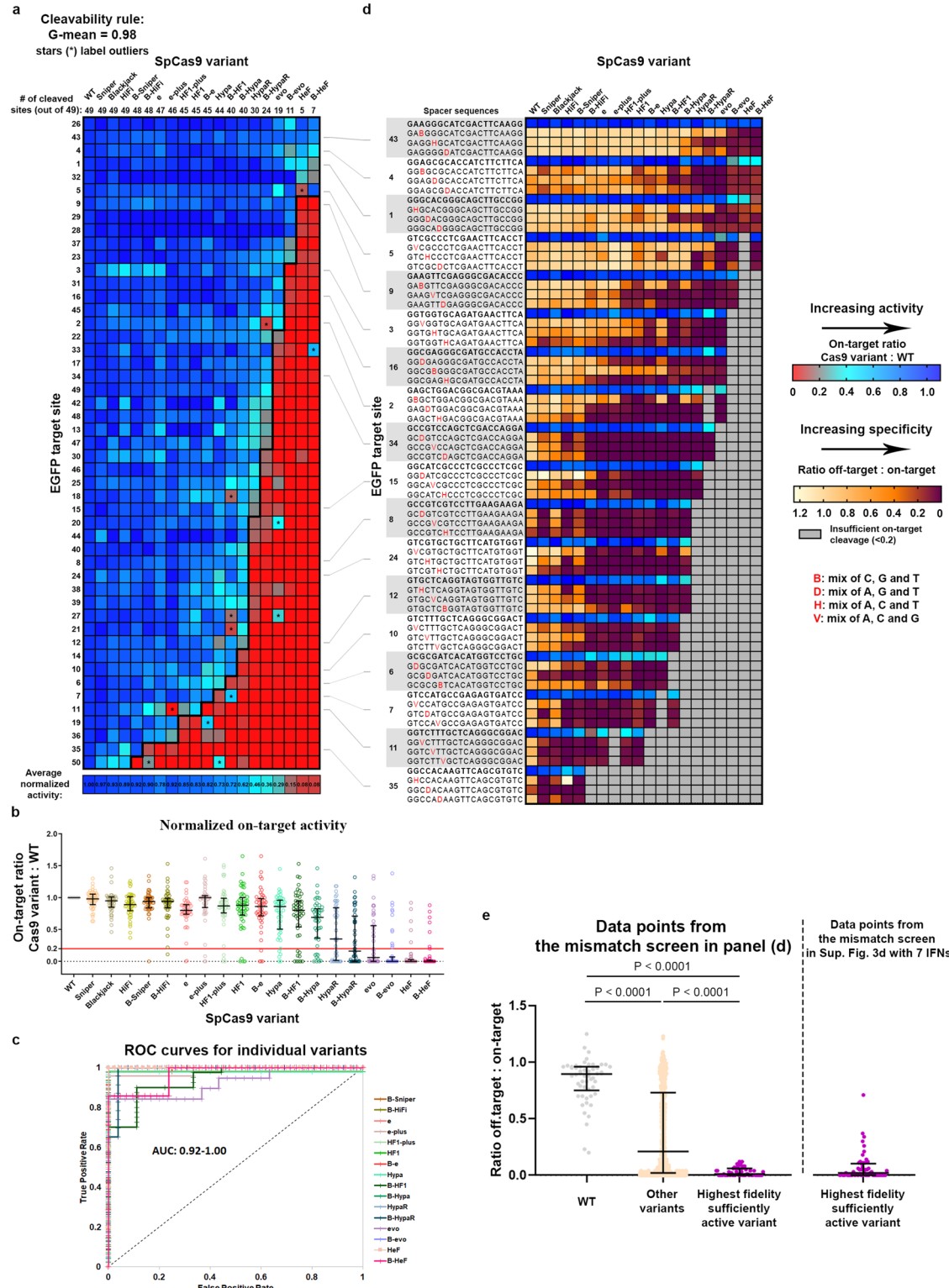

genome-wide off-target effects of an IFN correspond to its EGFP assay mismatch tolerance, both being primarily determined by the position of the target and the IFN within the two-dimensional ranking map. The number of off-targets in GUIDE-seq and the specificity of editing in the disruption assay change in parallel: decrease and increase, respectively. (ii) The fidelity rank of the IFNs and the cleavage rule remained in effect when cleavage was examined in different cell line (HEK293) and on 52 endogenous target sites (G-mean score of 1.00, Fig. 3a, b). (iii) We further verified these results on the largest possible target data

set available from the literature, where targets were examined with more than two IFNs. Kim et al. published the activity data of WT, Sniper, e-, evo-, Hypa- and SpCas9-HF1 on 6,481 target sequences that were suitable for our analyses[34]. These data confirmed the same activity/fidelity order of these five IFNs as we reported in this study. We analyzed these data in silico from more than 32,000 (5 × 6481) data points and found that these target sequences also tightly follow the cleavage rule (G-mean scores of 0.981, Fig. 3c–e). This study also provided off-target cleavage data, of which we analyzed the results

**Fig. 2 | Extending the set of IFNs enables increased specificity.** Heatmaps show the normalized EGFP disruption activity of SpCas9 nucleases with **a** perfectly matching and **d** partially mismatching 20G-sgRNAs in N2a cells. **a** The bold line indicates the dividing line defined by the cleavage rule between the classes of cleaved and not-cleaved values. The G-mean value indicates how well the data points above and below the bold line correspond to cleaved and not-cleaved (<0.20 activity normalized to WT) experimental values. Targets and IFNs are shown in the same order as in Supplementary Fig. 3. **b** Normalized on-target disruption activities of various SpCas9 variants presented on a scatter dot plot. The sample points correspond to data presented in (**a**) $n = 49$. The median and interquartile range are shown; data points are plotted as open circles representing the mean of biologically independent triplicates. Continuous red line indicates 0.20 normalized disruption activity, under which we consider the IFNs not to be active on a given target. Statistical significance was assessed by using RM one-way ANOVA and is shown in Supplementary Data file 9. **c** The ROC curves demonstrate that the order of the target sequences, determined by the cleavage rule, competently separates the classes of cleaved and not-cleaved normalized disruption values of each of the 19 variants from (**a**). **d** Mismatch screen of the nuclease variants either with perfectly matching 20G-sgRNAs or with mismatched sgRNAs (a mixture of three different sgRNAs used for each examined mismatch position[33]) as indicated in the figure. Gray boxes: not determined because on-target activity was too low. Targets from higher ranks (cleavable by many IFNs) require higher fidelity nucleases, while targets from lower ranks (cleavable by few IFNs) require lower fidelity nucleases for editing with both high efficiency and high specificity. **e** Matching IFNs to targets further increases the specificity of editing. The highest fidelity still active variants from the 19 IFNs in (**d**) provide more specific editing then those from the 7 IFNs shown at the right of the panel. The median and interquartile range of data points selected from (**d**) is presented as indicated; $n = 54, 654, 54$, respectively. Dots are shown for each variant with each mismatching spacer position, provided that the on-target activity exceeded 70%; data are omitted otherwise. Statistical significance was assessed by RM one-way ANOVA, statistical details and exact $p$-values are available in Methods and in Supplementary Data file 9. **a–e** Target sequences, raw and processed disruption data and statistical details are reported in Supplementary Data files 1–4, 9.

from 30 sgRNAs on perfectly matching target sequences along with 1800 off-target sequences containing all possible one-nucleotide mismatches for all nucleotide positions of the target and for all possible types of nucleotide change. The analyses confirmed our conclusion, that targets from different cleavability ranks require IFNs with correspondingly different fidelity ranks for specific cleavage (Fig. 3f). Here again, selecting the IFN that is closest to a target-matched one substantially increased the accuracy of IFN-editing (Fig. 3g). However, as seen in Fig. 3g, there are still considerable off-target effects remaining when using only these 5 IFNs. This is consistent with the idea that a larger number IFNs can ensure a better resolution, and thus, provide an appropriate fit for more targets from the same set of target sequences. All the above results demonstrate that the sequence contributions of the targets in combination with the effect of the fidelity-increasing mutations of the IFNs primarily regulate on-target and off-target cleavage. These features appear to be universal; not specific to one cell-type or assay-type, and it applies to all variants and targets tested.

**The cleavage rule is discernible in the in vitro activities of IFNs**

Next, we investigated whether the sequence contributions of the targets directly affect the cleavage activity of the variants, or they may derive solely from cellular effects. 21 targets from various cleavability ranks from Fig. 2 were examined in an in vitro plasmid cleavage assay employing the purified ribonucleoprotein (RNP) complex of the WT SpCas9 and of either B-SpCas9-HF1, a variant from the middle of the fidelity ranking, or B-evoSpCas9 from the higher ranks (Fig. 4 and Supplementary Fig. 6). Figure 4c reveals that target sequences impact the activity of SpCas9s differently yielding a more than a magnitude difference in the cleavage rates in case of each variant, consistent with an earlier report[45]. Fidelity-increasing mutations decrease the activity of B-evoSpCas9 more than that of B-SpCas9-HF1 in a target-dependent manner. Most importantly, Fig. 4c shows that the combined effect of target sequence contributions and fidelity-increasing mutations is not only apparent in cellulo, but also in vitro, therefore it directly affects the cleavage activity of SpCas9s.

Two other arguments also support indirectly that the cleavage rule results from a direct interaction between IFNs and targets. (i) The EGFP disruption experiments demonstrate that the observed differences in the cleavability of the targets by IFNs (but not WT) in cellulo does not result from the location of the targets (whether in chromatin, coding or non-coding regions) or from the transcript levels when they are in a transcribed region, since the targets shown in Fig. 2a are all located within the EGFP sequence integrated into one location of the genome. (ii) It could also be argued that the cleavage pattern seen in the heatmap in Fig. 2 might be the result of the IFN expression levels, or alternatively, higher-ranking IFNs may have WT-

like activity at high-ranking targets only because cleavage at these targets is saturated, and therefore their reduced activity is not apparent. Hence, we made all reasonable efforts to ensure identical expression levels of the IFNs; they were expressed from the same vector with identical codon optimization, differing only at the mutated positions. Transfection efficiency was monitored with a fluorescent marker for both EGFP disruption and NGS amplicon sequencing experiments. Also, to examine whether cleavage was saturated, we performed titration of plasmids expressing higher ranking IFNs from the disruption assay with a few targets that had been either cleaved or not cleaved by the IFNs in Fig. 2. Supplementary Fig. 7 shows that the system is saturated for both the WT and the IFN proteins. With certain targets, the activity of variants with less amount of plasmid starts declining sooner than the activity of the WT, resulting in these variants having a reduced activity on these targets compared to the WT. However, this reduced activity is markedly different from the almost complete loss of activity of the variant that should not cut the target according to Fig. 2, suggesting that saturation is not the cause of the observed pattern.

Using the variants in a pre-assembled RNP form, the rank order of IFNs and targets was reproduced with a single outlier out of the 99 cleavages (G-mean score of 0.987, Supplementary Fig. 8a, b). As expected, the IFNs in pre-assembled RNP form showed lower activities and frequently increased specificities while preserving the characteristics of the cleavage rule demonstrated with plasmid transfection (Supplementary Fig. 8c–f).

Taken together, these data suggest that the combined direct effect of target sequence contributions, fidelity-increasing mutations and mismatches on SpCas9 activity result in the emergence of the cleavage rule. When the activating effect of target sequence contributions is much larger than the effect of fidelity-increasing mutations, then not only target cleavage occurs with substantial off-target effects, but also the impact of other intrinsic and cellular factors is more pronounced, modulating the level of WT-normalized activity, typically between 70% and 120% (Fig. 2a).

**SpCas9-NG and xCas9 do not obey the cleavage rule**

With the established knowledge that an appropriate set of IFNs rather than any individual variant is necessary for reaching maximal specificity universally for any target, it would be a particularly useful idea to create an alternative set of IFNs with altered-PAM specificities. This would increase the accessibility of target sequences by the recognition of targets with an NG-like PAM sequence instead of the canonical NGG. Such variants, like SpCas9-NG and xCas9, have also been reported to possess increased fidelity and relatively low activity[46,47]. In order to create IFNs that belong to the lower fidelity ranks but with NG PAM specificity, the activity of the SpCas9-NG or xCas9 would need to be

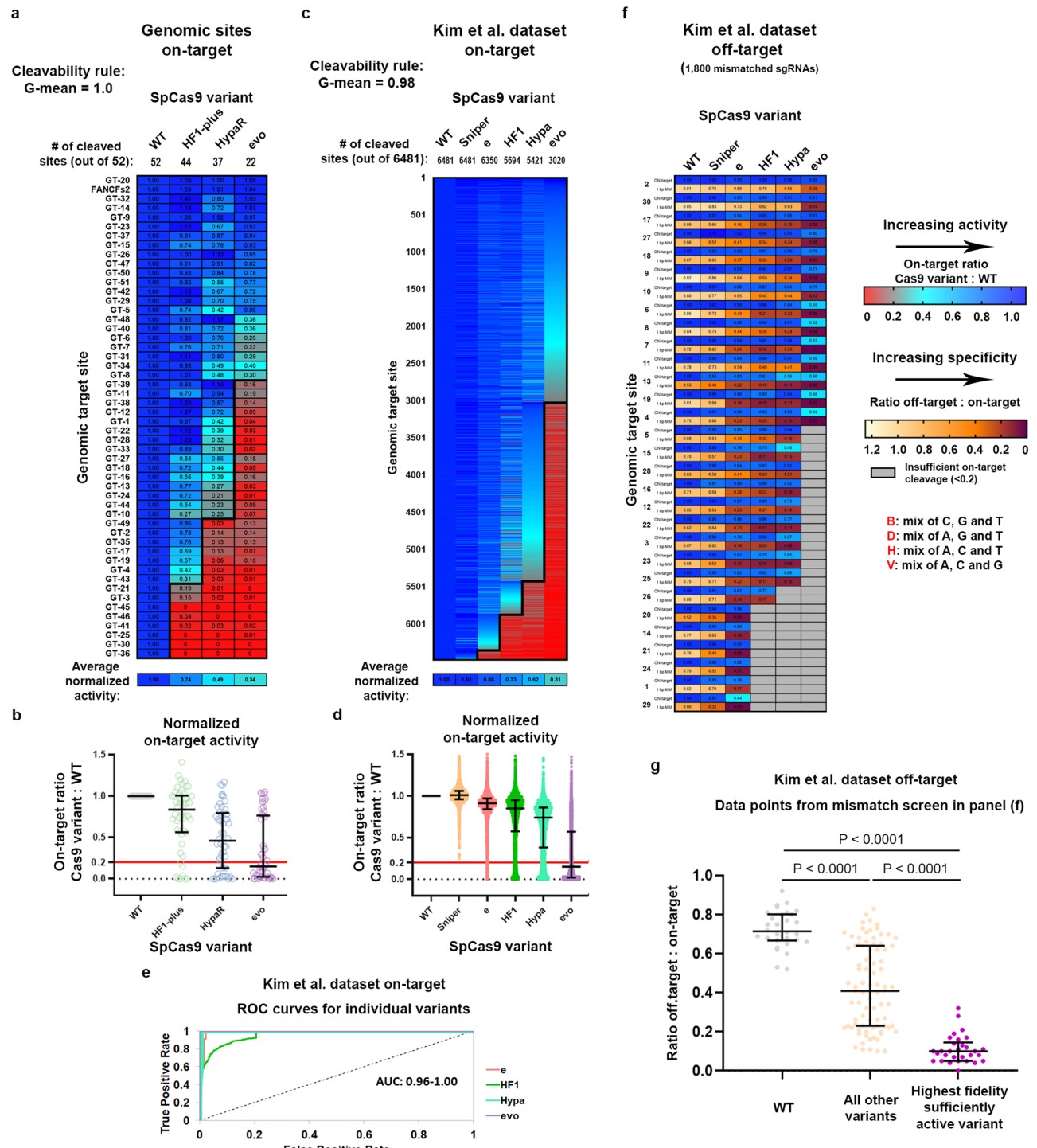

increased. Some mutations in xCas9 have been hypothesized to primarily increase the fidelity of the variant, instead of contributing to the altering of the PAM specificity[48]. Our efforts to increase its activity by replacing these mutations with the wild type amino acids were unsuccessful (Fig. 5a). As an alternative solution, we applied fidelity decreasing mutations[22] and demonstrated their effects on the activity of HypaR-SpCas9, an IFN with activity close to that of SpCas9-NG and xCas9, and on targets that are in the cleavability ranks just on the border of cleaved or not cleaved by HypaR. However, when we introduced them to SpCas9-NG or xCas9, the mutations did not decrease their target selectivity on the tested sequences (Fig. 5b, c). Intriguingly, SpCas9-NG and xCas9 do not fit in with the pattern formed by the rest of the IFNs (Fig. 5d). They do not strictly obey the cleavage rule of the

targets (Fig. 5e). In this respect, it would be interesting to see other PAM-altered variants, such as ones developed by Kleinstiver and co-workers, whether they also behave like SpCas9-NG and xCas9[49–54]. These results also highlight, that a variant with reduced activity, even with seemingly increased specificity, does not automatically qualify for the IFN ranking, and that the cleavage rule resulting the pattern seen in Fig. 2 is not something self-evident.

From here, we progressed parallel, on the one hand, (i) with the generation of more IFNs with intermediate fidelity for ranks with lower resolution, while on the other hand, (ii) proceeding with this set of 19 variants to assess if this panel was large enough to demonstrate that target-matched IFNs facilitate genome editing with maximal specificity i.e., without any detectable genome-wide off-target.

**Fig. 3 | The cleavage rule still applies in a different cell line (HEK293) and on endogenous target sites examined by NGS.** Heatmaps show the normalized value of the percentage of genome modification induced by SpCas9 variants with **a, c** perfectly matching and **e** partially mismatching 20G-sgRNAs. **a, c** The bold line indicates the dividing line defined by the cleavage rule between the classes of cleaved and not-cleaved values. The G-mean value indicates how well the data points above and below the bold line correspond to the cleaved and not-cleaved (<0.20 activity normalized to WT) experimental values. **b, d** Normalized on-target genome modification rates of various SpCas9 variants presented on a scatter dot plot. The sample points correspond to data presented in (**a**) and (**c**); $n = 52, 6481$, respectively. Continuous red line indicates 0.20 normalized disruption activity, under which we consider the IFNs not to be active on a given target. The median and interquartile range are shown; data points are plotted as open circles representing the mean of biologically independent triplicates. Statistical significance was assessed by Friedman test and is shown in Supplementary Data file 9. **c, d** The data are compiled from experiments from Kim et al.[34] and contain only selected sequences to avoid the effect of 5′ extended sgRNAs known to diminish the activity of all IFNs except Sniper and the SpCas9 variants containing the Blackjack mutations[21,27] (details can be found in Materials and Methods section and in Supplementary Data file 7). The G-mean score of **a** 1.00 and **c** 0.98 (only 171 out of 32,405 data points are outliers) confirms that the cleavage rule is the main factor determining the activity of IFNs on genomic target sequences. **e** The ROC curves verify that the order of the target sequences, determined by the cleavage rule, competently separates the classes of the cleaved and not-cleaved targets based on the normalized genome modification values of each individual variant from (**c**). **f** Mismatch screen of six nuclease variants with 30 sgRNAs on either perfectly matching or one-base mismatching target sequences. Data are compiled from experiments from Kim et al.[34] and contain outcomes on 1,800 off-target sequences (details can be found in Materials and Methods section and in Supplementary Data file 7). **g** Matching IFNs and targets further increases the specificity of editing. The median and interquartile range of data points that are selected from (**f**) is presented as indicated; $n = 30, 87, 30$, respectively. Dots are shown for each variant and target pair, where the on-target activity exceeded 70%. Statistical significance was assessed by RM one-way ANOVA, statistical details and $p$-values are available in Methods and in Supplementary Data file 9. **a–g** Target sequences, heatmap, NGS and data selected from Kim et al. and statistical details are reported in Supplementary Data files 1, 5, 7, 9.

## Extending the set of IFNs with ten additional variants with intermediate fidelity

One of the main conclusions of our study is that a full series of IFNs is needed in order to be able to provide highly specific editing in general, for any given target regardless of its cleavability rank. However, the distribution of our set of 19 increased-fidelity SpCas9s is not spread evenly across the full range of the fidelity ranking. There are more IFNs in the lower/medium fidelity range and some of them do not or just marginally differ in on-target activity/fidelity. In contrast, there are only a few options for targets requiring nucleases with higher fidelity. Therefore, to provide a better resolution of the available IFNs in these higher fidelity ranks, we reverted several single mutations[22] in B-evo- and B-HeFSpCas9 to the original WT amino acids creating ten additional variants with the intended intermediate fidelity and target-selectivity (Fig. 5f). These variants provide additional tools for editing those targets from the high cleavability ranks where the panel of the 19 variants may not provide a sufficiently matching IFN.

## Identifying the target-matched variants

Finally, the most important result of this study is that by employing target-matched IFNs we are able to ensure maximal specificity editing for practically any target sequence, that is accessible to WT SpCas9, without any genome-wide off-target effects. Several clever and effective genome-wide off-target detecting methods have been developed in vitro and in cellulo[11,20,55–67]. While in vitro methods tend to report more off-target sites, they are prone to identifying a high number hits with uncertain relevance and require extensive validations. In addition, the off-target sites reported exclusively by in vitro methods, such as Digenome-seq or CIRCLE-seq, are typically amongst the minor off-target events. The major off-target cleavage events are usually reported by both GUIDE-seq and in vitro approaches[55], and these are the last remaining ones that target-matched variants should eliminate. Thus, the minor off-target events do not seem to be relevant in our experiments. In addition, since in vitro methods require validation by amplicon sequencing, their detection limit here is determined by the sensitivity of the NGS in the amplicon sequencing. As opposed to in vitro methods, GUIDE-seq, likely the most widely used approach, is reported to have the highest validation rate amongst genome-wide methods[58] and its sensitivity is comparable to or, with certain targets, even higher than that of amplicon sequencing[68,69]. Thus, given the rather large number of pairs of target and variant to be tested, in this study we relied on GUIDE-seq to monitor the off-target activity of the nucleases, backed by NGS validation for the top three sites identified by GUIDE-seq). To identify the target-matched variants for a given target and to select the optimal one for maximum specificity without having to test all of the IFNs in the set, we used a two-step method by exploiting the observed cleavage rule of the targets. To reduce the number of variants to be tested, we omitted two IFNs from the low fidelity range of the IFN set shown in Fig. 2, where the fidelity of IFNs differs very little from each other. We refer to these remaining 17 IFNs as CRISPRecise set. The schematic of the method is demonstrated on a hypothetical target example (Fig. 6a). In the first step, we measure the on-target activity of WT and three IFNs (e-plus, B- HF1 and B-HypaR), that divide the target range in Fig. 2a into four proportional sections based on the fraction of the targets they can cleave, to identify which one has the highest fidelity, that still shows sufficient efficiency. In the second step, a few additional IFNs, between the last working and the first non-working (identified in the first step), are tested for on-target activity to identify target-matched variants. Finally, out of these variants, the optimal, target-matched variant with the maximum specificity is selected and/or confirmed by GUIDE-seq. We show this strategy on HEK site 1, 2, 3 that have been analyzed by GUIDE-seq previously[20]. Figure 6b demonstrates that with the CRISPRecise set, all three targets could be edited without any genome-wide off-target effect detected by GUIDE-seq.

## Editing challenging targets efficiently without any detectable off-targets

We have shown the usefulness of the application of target-matched IFNs on 3 genomic targets (Fig. 6b), however, in order to draw meaningful conclusions, instead of simply adding extra arbitrarily selected targets, we challenged our approach by examining all problematic target sequences reported in the literature that have been failed to be edited by any of the IFNs without genome-wide off-targets (Supplementary Data file 6: Data from other studies)[20,21,23,24,26,28]. Most studies characterizing IFNs focused on the same or an overlapping set of targets in order to provide the new variant with a relevant comparison to the preceding ones. Thus, these studies together ended up examining the same targets with a number of variants and by chance some of these tests involved a target-matched variant for several of the targets. In some cases, the target could not be edited without off-targets in spite of all efforts, simply because the existing/tested variant IFN set did not contain the target-matched variants. Here we tested our approach on the eight targets on which the former IFN-studies failed to provide efficient and off-target-free editing[21,23,24,27,29]. In contrast to previous studies, using both the understanding of the cleavage rule and the extended set of IFNs (CRISPRecise set) developed here, we identified the target-matched variants and managed to successfully edit all eight challenging targets without any GUIDE-seq-detectable genome-wide off-target (Fig. 7 and Supplementary Fig. 9b–f).

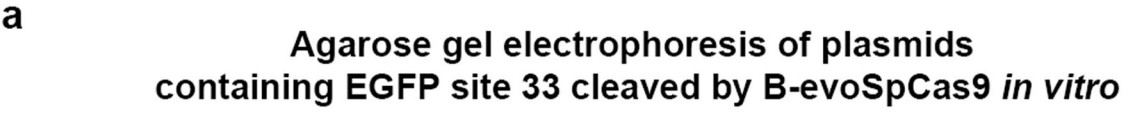

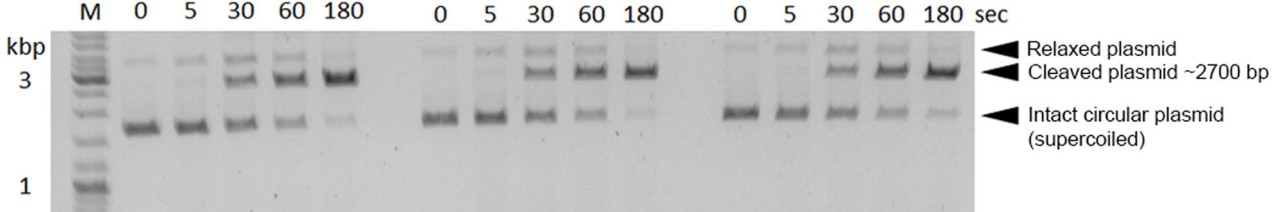

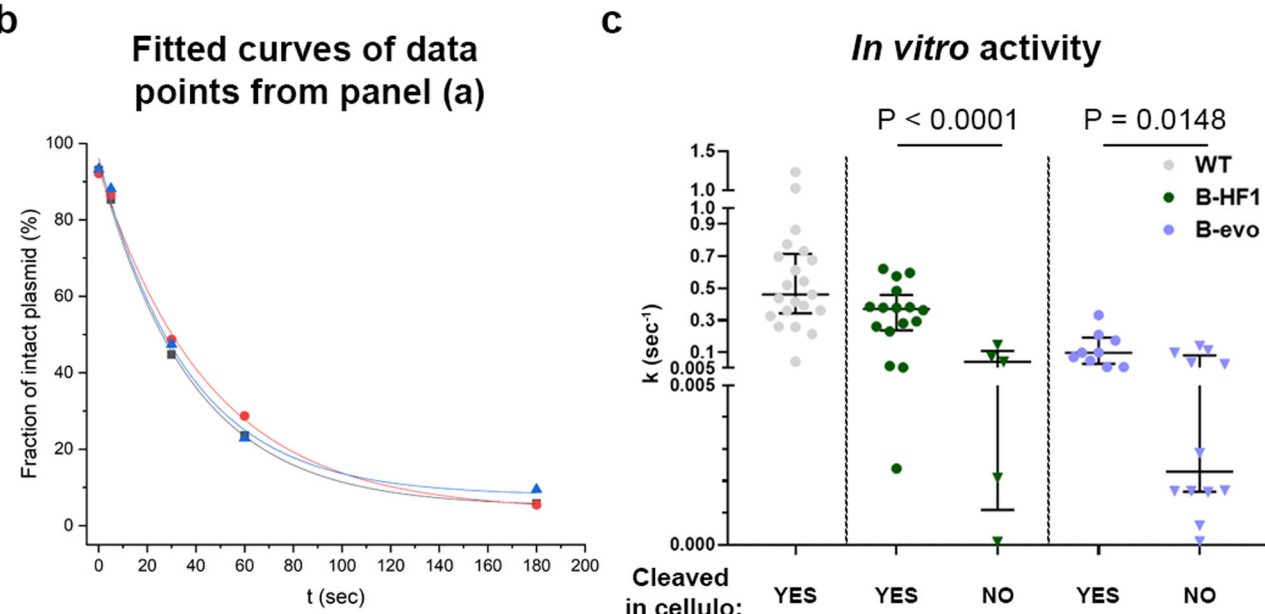

**Fig. 4 | Sequence contributions are evident in vitro, supporting that it directly affects the enzymatic activity of IFNs.** The effect of increased-fidelity mutations and sequence contributions seen in cellulo also manifests in the rate constants of in vitro cleavage activities of the variants employing 21 targets of Fig. 2. **a** Representative agarose gel showing the activity of B-evoSpCas9 variant in a plasmid-cleaving in vitro assay at different timepoints in triplicates. **b** Plot showing values representing the consumption of the intact circular plasmid (not-cleaved) derived from the intensity of bands from the representative agarose gel in (**a**). Exponential curves were fitted to the timepoints of each replicate separately. **c** The average k values of three individual fits are shown for 21 EGFP target sites, separated into

categories based on the in cellulo cleavage results. The median and interquartile range are shown; data points represent the mean of the fitted k value triplicates; $n = 21, 16, 5, 9, 12$, respectively. Differences between groups were tested by using either two-tailed unpaired Student's t-test with Welch's correction (B-SpCas9-HF1) or by using two-tailed Mann–Whitney test (B-evoSpCas9) in the cases where differences did not meet the assumptions of unpaired t-test. Statistical details and exact p-values are available in Materials and Methods section and in Supplementary Data file 9. **a**–**c** Data related to Supplementary Fig. 6. Target and primer sequences, in vitro data and statistical details are reported in Supplementary Data files 1, 8, 9.

Interestingly, amplicon sequencing revealed no off-target sites in any of the target-IFN pairs, except for one where GUIDE-seq detected none, while in another case amplicon sequencing detected no off-target modifications whereas GUIDE-seq found reads with one (*VEGFA* site 1 evoSpCas9 RNP; Supplementary Fig. 10). In the former case, B-HeFSpCas9 seems to have a small residual off-target effect with target *CCR5* site 11 (Supplementary Fig. 10). This target has the highest cleavability in our study, indicating that additional IFNs with higher fidelity than the existing ones should be developed to address such rare, high cleavability targets. Most impressively, *VEGFA* site 2, 3 and *FANCF* site 2, which have been previously failed by 7, 4 and 7 IFNs, respectively[21,23,24,26], were also edited without genome-wide off-targets by using target-matched nucleases in RNP form. These results project that by the use of an appropriate set of IFNs virtually any target from any rank can be edited with greatly enhanced specificity, without any off-target effect (experiments summarized in Fig. 8 and detailed in Figs. 6, 7, 9, Supplementary Figs. 4, 5, 9–11). The greatest benefit of

these results is likely to be realized in therapeutic applications of genetic engineering, where maximum specificity and safety are required.

## Correcting a clinically relevant mutation without any detectable off-target

We also attempted to correct a clinically relevant mutation in a patient-derived cell line to present the power of the method on a relevant target site that we had no prior knowledge of. Cells with a defective mutation were derived from a patient with Xeroderma pigmentosum, a rare genetic disorder without any cure to date[70]. They harbor a C>T substitution, which results in the change of Arg-683 to Trp disrupting the function of the *ERCC2* gene. Patients with Xeroderma pigmentosum are extremely sensitive to the ultraviolet range of sunlight as a result of dysfunctional DNA repair, which often leads to the development of skin cancer and early death at a young age[71]. We located the target sequence nearest to the mutation, identified the optimal target-

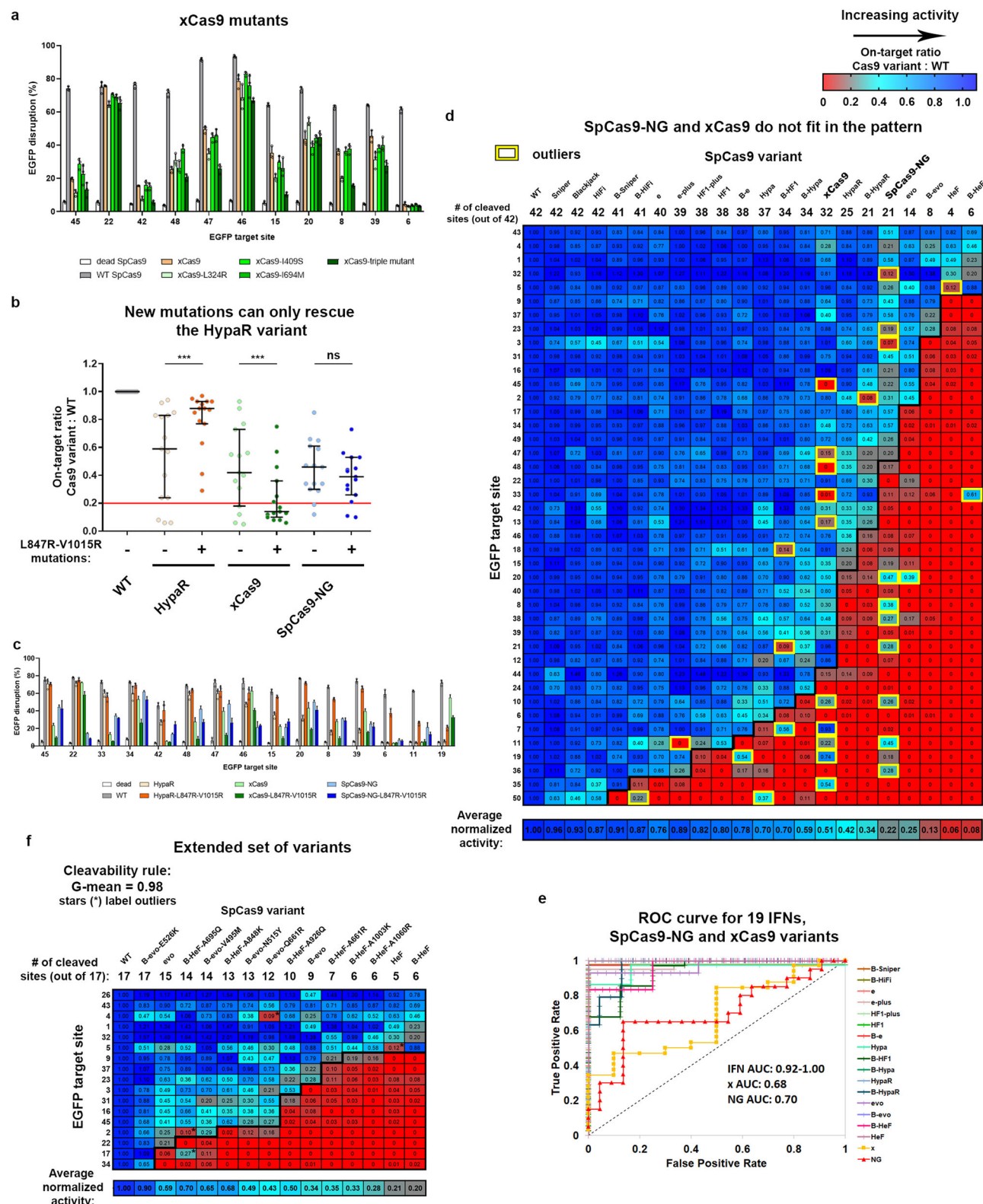

matched IFN and corrected the mutation with B-HypaSpCas9 in 10.7% of the cells using single-stranded DNA oligonucleotides without any detectable off-target effect, indicating the high potential of our approach (Fig. 9).

## Discussion

There are two major achievements in this study; on the one hand we recognized the cleavability ranking of the targets and established the cleavage rule, that governs the outcome of the interactions between IFNs and targets, and then, used this knowledge to develop additional IFNs that fill the gaps in the fidelity ranking of the variants so that we may provide a suitable IFN for targets from any cleavability rank. On the other hand, by exploiting the cleavage rule and the additionally developed set of IFNs we demonstrated that both maximal specificity (i.e., no detectable genome-wide off-targets, as assessed by GUIDE-seq) and efficient cleavage can be expected to be achieved universally

**Fig. 5 | Increased-fidelity SpCas9 variants in the higher fidelity ranks with in-between activity/fidelity. a, c** The results of an on-target EGFP disruption assay for WT and different SpCas9 variants on various target sites shown on a column graph. Means and SD are shown; *n* = 3 biologically independent samples (overlaid as white circles). **a**, Reverting three mutations of xCas9 back to the WT residues, suggested by Guo et al.[48] as being responsible for its increased fidelity and target-selectivity, did not increase its activity the way we expected. Statistical significance was assessed by RM one-way ANOVA and shown in Supplementary Data file 9. **b** Normalized on-target activities of various SpCas9 variants presented on a scatter dot plot. The sample points correspond to data presented in (**c**); *n* = 14. Continuous red line indicates 0.20 normalized disruption activity, under which we consider the IFNs not to be active on a given target. The median and interquartile range are shown; data points are plotted as open circles representing the mean of biologically independent triplicates. Differences between SpCas9 variants were tested by using either two-tailed paired-samples Student's *t*-test or by using two-tailed Wilcoxon Signed Ranks test in the cases where differences did not meet the assumptions of Paired t-test. Statistical details and *p*-values are available in Methods and in Supplementary Data file 9. **d, f** Heatmaps show the normalized EGFP disruption activities of SpCas9 nucleases with perfectly matching 20G-sgRNAs. The bold line indicates the dividing line defined by the cleavage rule between the classes of cleaved and not-cleaved values. **d** SpCas9-NG and xCas9 do not strictly obey the cleavage rule when fitted on the heatmap of Fig. 2a (only 42 EGFP target sites were tested here), even though some of the targets for which the order were not determined by the 19 IFNs were reordered (compared to Fig. 2a heatmap) to favor the accommodation of SpCas9-NG and xCas9 into the cleavage map. These results explain the failure to develop an IFN series with a looser PAM requirement and emphasize that the cleavage rule identified here is non-trivial and does not apply to all other SpCas9 variants with reduced activity or increased fidelity. **e** The ROC curves demonstrate that the order of the target sequences, determined by the cleavage rule, competently separates the classes of the cleaved and not-cleaved targets in case of the IFN variants, but xCas9 (AUC: 0.68) and SpCas9-NG (AUC:0.70) do not appear to strictly follow the cleavage rule, emphasizing that the rule is not self-evident. **f** Additional IFNs provide a finer resolution within the higher fidelity region of the IFN ranking between evo- and HeFSpCas9, and their activities on these targets also strictly follow the cleavage rule. For these experiments, we selected targets from the higher cleavability region of the target ranking in Fig. 2, as these were expected to be the point of distinction between the additional variants, and therefore facilitate the ordering of these IFNs and their fitting into the already existing ranking. This is the reason for these high fidelity IFNs showing much higher normalized disruption activities, than they would on randomly selected targets. **a–f** Target sequences, raw, processed, heatmap disruption data and statistical details are reported in Supplementary Data files 1–4, 9.

for any target. We note five issues related to the cleavage rule; (i) The cleavage rule perfectly separates IFNs into cleaving and non-cleaving groups for a specific target (G-means range between 0.98 and 1.0 in our results, Figs. 2, 3 and Supplementary Fig. 3), but it does not necessarily mean that cleaving IFNs show continuously decreasing normalized activities on a given target according to their ranking. Their WT-normalized activities typically scatter between 75 and 125%. This is likely because at the point, when target sequence contribution has already ensured effective cleavage, there is no room for further improvement by facilitating the docking of the HNH domain, since the HNH domain had already been stably docked in active conformation. However, other factors that exert their effects on modulating the activities of these cleaving IFNs in a different way may become apparent. For the same reasons, target contributions are also less evident in the WT cleavage pattern. (ii) The recognition of the cleavability ranking of SpCas9 targets may inspire researchers to revisit some structural and mechanistical studies of SpCas9, that are typically performed on a single target, by examining targets of different ranks to cross-check their conclusions. (iii) This knowledge is particularly important for studies where a selection scheme[26,27,29,30] is set up with a single target, which then only allows the development of IFNs whose activity is limited by the cleavability ranking of the target.(iv) Efforts to engineer a variant with significantly increased fidelity without compromising activity have been unsuccessful[72]. Our results suggest that this can only be achieved with a mutant variant that is activated by all target sequences to approximately the same extent. (v) Furthermore, in vitro data confirmed that sequence contributions resulting in the cleavability ranking of the targets directly affect the cleavage activity of these SpCas9 nuclease variants, however, further research is required to understand what sequence features exactly are at work.

Regarding of the use of target-matched IFNs, we highlight the following. (i) The larger the IFN set that is being used to identify the highest ranking IFN with sufficient activity, the more likely it is to contain the IFN with maximum specificity to the target. Using the 3 IFNs (from Set A) obtained from the first rough screen, we could edit with a largely increased specificity, although for most targets some off-target modifications will still be detectable. Using Set B, we could achieve maximum specificity for a significant proportion of the targets. Actually, all but one of the targets examined here could be edited without any genome-wide off-target modifications by using IFNs selected from Set B. (Table 1). (ii) Although great improvement in fidelity can be achieved with little effort by using just the three IFNs from the first screening step, the target-matched nuclease obtained from the two-step screening process provides highly specific and efficient editing. When achieving maximum specificity is critical, it may be wiser to confirm maximal specificity by testing the two best candidates with a genome-wide off-target detection method, as the activity of an IFN is influenced by a number of factors, leading to, in some infrequent cases, unexpected outliers with residual off-target effects. In this study we found only one case where a variant, which ranked lower than the target matched variant, had maximum fidelity, unlike the target matched variant. (iii) Here, we showed that practically any target that is efficiently edited by the WT SpCas9 can be expected to be edited efficiently, without off-targets by employing target-matched IFNs, thus considerably increasing the potential of genome engineering in terms of safety and efficiency when high specificity is required, such as gene therapeutics. (iv) In gene therapeutics, although the majority of the off-target mutations may have no detrimental consequences, the few that do still uphold substantial threat as ex vivo and in vivo therapeutic applications involve millions to billions of cells. The routine use of a given therapy further increases the risk by thousands of folds, in contrast to a single treatment. Furthermore, the off-target cleavages by the nuclease even in innocuous positions can still pose a significant risk, as double-strand breaks at off-target positions increase the chance of chromosomal translocations that can also lead to cancerous transformation[13,73]. (v) For safe therapeutic procedure the aim needs to be maximal specificity, possibly beyond the about 0.1% detection limits of current methods[58,74] for the assessment of off-targets. Since a target may be edited without detectable off-targets by multiple IFNs, in such cases, as a general practice, the target-matched IFN with the highest fidelity should be identified and applied. RNP form delivery has been shown to preserve the fidelity order of the IFNs and the cleavability order of the target, however the highest fidelity variant showing sufficient activity may be different due to the shorter and lower level presence of the variants in the cells. To further increase specificity, the lower fidelity neighbors and the target-matched variant may also be tested with other fidelity-enhancing approaches such as RNP form or dRNA[75], and it is worth considering their application to maximize specificity even in cases that would fall under the detection limits of off-target detecting methods. (vi) The use of target-matched IFNs may also be beneficial in base and prime editing[76–78]. These methods work with substantially less Cas9 dependent off-targets than nucleases, nevertheless, they also rely on cleavage, i.e., the nickase activity of SpCas9. The nickase versions of IFNs seem to exhibit the same sensitivity to the sequence contributions of the targets[32,79], thus applying target-matched IFN base and prime editors may decrease off-

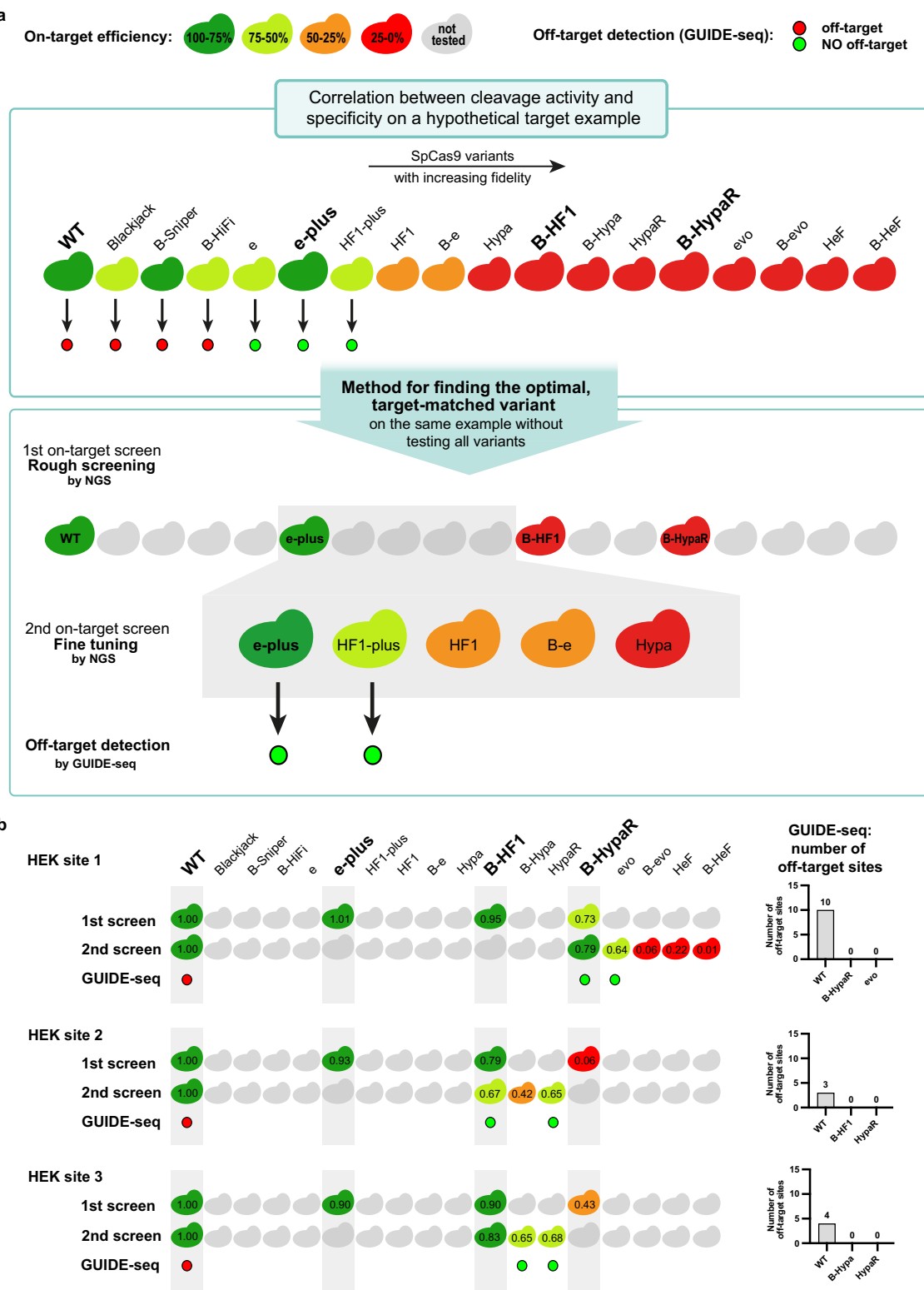

target editing of current editors to a non-detectable level and further. (vii) Here, the identification of target-matched IFNs for a given target has proved to be relatively straightforward, still, a predictive algorithm, which could identify the target-matched IFNs for specific targets could further simplify this process and make it less labor intensive. Unfortunately, prediction programs to date are not accurate enough to suggest a reliable choice (The specificity of the predictions is ≤0.5 for all IFNs that can be tested from Fig. 2a using either DeepSpCas9 or DeepRank that we developed in this study using a subset of the data

generated in ref. 34 see Supplementary Table 1). Large cleavage activity data for a considerable number of IFNs from all fidelity ranges of the ranking should be generated for the development of an appropriate prediction tool.

In conclusion, the translation of advances in CRISPR technology into clinical applications faces several challenges in terms of the efficiency of the modification, the delivery of the tools in vivo as well as various undesired, non-intended modifications affecting the genome. Our approach substantially diminishes one of these obstacles; the

**Fig. 6 | The optimal, target-matched SpCas9 nuclease, which shows efficient on-target editing and no off-target effects, is identified for each target using a two-step approach. a** Schematic representation of the two-step screening method used on a hypothetical target example. The first panel shows the on- and off-target activity of a set of IFNs with increasing fidelity on a hypothetical target example. The second panel shows the screening method, which identifies the optimal variant for the target without having to test all of the variants. In the first step, a rough on-target screen is performed, where the WT and three selected IFNs, that divide the target ranking range into four approximately proportional sections, are tested. The second step is a fine-tuning on-target screen, that involves the not yet used variants with higher fidelity than the highest ranking active (green) variant from the first screen, and it identifies the target-matched variants (active variants with the highest fidelity). If necessary, two sufficiently active (here their normalized activity is above 50%, but this may depend on the application under consideration) target-matched variants can be screened for the absence of genome-wide off-targets. **b** The identification of the target-matched variants that provide appropriate editing without any genome-wide off-target is demonstrated on three targets that had been tested in Tsai et al.[20]. The numbers in the colored Cas protein illustrations indicate the percentage value of the on-target genome modifications normalized to WT (measured by NGS). Colored circles indicate whether a target was edited with (red) or without (green) off-targets in the GUIDE-seq experiment. The total number of off-target sites detected by GUIDE-seq are shown for each target in bar charts on the right side of the panel. Data related to Fig. 8, Supplementary Figs. 10, 11. Target sequences, NGS and GUIDE-seq data are reported in Supplementary Data files 1, 5, 6.

appearance of off-target edits, and therefore it provides an exceptionally high precision tool for research and therapeutic applications.

# Methods

## Materials
Restriction enzymes, T4 ligase, Dulbecco's modified Eagle Medium DMEM (Gibco), fetal bovine serum (Gibco), Turbofect, TranscriptAid T7 High Yield Transcription Kit, Qubit dsDNA HS Assay Kit, Taq DNA polymerase (recombinant), Platinum Taq DNA polymerase, 0.45 μm sterile filters and penicillin/streptomycin were purchased from Thermo Fischer Scientific, protease inhibitor cocktail was purchased from Roche Diagnostics. DNA oligonucleotides, trimethoprim (TMP), chloroquine, polybrene, puromycin, calcium-phosphate and GenElute HP Plasmid Miniprep kit were acquired from Sigma-Aldrich. ZymoPure Plasmid Midiprep kit and RNA Clean & Concentrator kit were purchased from Zymo Research. NEBuilder HiFi DNA Assembly Master Mix and Q5 High-Fidelity DNA Polymerase were obtained from New England Biolabs Inc. NucleoSpin Gel and PCR Clean-up kit was purchased from Macherey-Nagel. Two millimeter electroporation cuvettes was acquired from Cell Projects Ltd, SF Cell Line 4D-Nucleofector X Kit S were purchased from Lonza, Bioruptor 0.5 ml Microtubes for DNA Shearing from Diagenode. Agencourt AMPure XP beads were purchased from Beckman Coulter. T4 DNA ligase (for GUIDE-seq) and end-repair mix were acquired from Enzymatics. KAPA universal qPCR Master Mix was purchased from KAPA Biosystems.

## Plasmid construction
Vectors were constructed using standard molecular biology techniques including the one-pot cloning method[80], *Escherichia coli* DH5α-mediated DNA assembly method[81], NEBuilder HiFi DNA Assembly and Body Double cloning method[82]. All SpCas9 variants were codon optimized the same way. Plasmids were transformed into NEB Stable competent cells or DH5alpha. For detailed cloning and sequence information see Supplementary Notes. A list of sgRNA target sites, mismatching sgRNA sequences and plasmid constructs used in this study are available in Supplementary Data file 1. The sequences of all plasmid constructs were confirmed by Sanger sequencing (Microsynth AG).

Plasmids acquired from the non-profit plasmid distribution service Addgene (http://www.addgene.org/) are the following:
pX330-U6-Chimeric_BB-CBh-hSpCas9 (Addgene #42230)[6], eSpCas9(1.1) (Addgene # 71814)[22], VP12 (Addgene #72247)[23], pMJ806 (#39312)[7], pBMN DHFR(DD)-YFP (#29325)[83] and p3s-Sniper-Cas9 (#113912)[27]. pX330-SpCas9-NG (#117919) was a kind gift from Hiroshi Nishimasu.

Plasmids developed by us in this study and deposited at Addgene are the following:
Expression plasmids for human codon-optimized increased-fidelity (i.e. high-fidelity) SpCas9 variants: B-Sniper SpCas9 (#207361), B-HiFi SpCas9 (#207362) HypaR-SpCas9 (Addgene #126757), B-HypaR-SpCas9 (Addgene #126764, B-evoSpCas9-V495M (#207363), B-

evoSpCas9- N515Y (#207364), B-evoSpCas9- E526K (#207365), B-evoSpCas9- Q661R (#207366), B-HeFSpCas9-A661R (#207367), B-HeFSpCas9- A695Q (#207368), B-HeFSpCas9-A848K (#207369), B-HeFSpCas9-A926Q (#207370), B-HeFSpCas9-A1003K (#207371), B-HeFSpCas9-A1060R (#207372).

Expression of increased-fidelity (i.e. high-fidelity) SpCas9 variants in bacterial cells: WT SpCas9 (#207373), Sniper SpCas9 (#207374), Blackjack SpCas9 (#207375), HiFi SpCas9 (#207376), B-Sniper SpCas9 (#207377), B-HiFi SpCas9 (#207378), eSpCas9 (#207379), eSpCas9-plus (#207380), SpCas9-HF1-plus (#207381), SpCas9-HF1 (#207382), B-eSpCas9 (#207383), HypaSpCas9 (#207384), B-SpCas9-HF1 (#207385), B-HypaSpCas9 (#207386), HypaR-SpCas9 (#207387), B-HypaR-SpCas9 (#207388), evoSpCas9 (#207389), B-evoSpCas9 (#207390), HeFSpCas9 (#207391), B-HeFSpCas9 (#207392).

The larger the IFN set that is being used to identify the highest ranking IFN with sufficient activity, the more likely it is to contain the IFN with maximum specificity to the target. Using the 3 IFNs (from Set A) obtained from the first rough screen, we could edit with a largely increased specificity, although for most targets some off-target modifications will still be detectable. Using Set B, we could achieve maximum specificity for a significant proportion of the targets. Actually, all but one of the targets examined here could be edited without any genome-wide off-target modifications by using IFNs selected from Set B. The CRISPRecise set (Set C), which includes all variants of Set A and B plus additional variants, allows editing practically with all target sites without any off-target effect, is available from Addgene as a plasmid kit (CRISPRecise kit) (Table 1).

## In vitro transcription
sgRNAs were transcribed in vitro using TranscriptAid T7 High Yield Transcription Kit and PCR-generated double-stranded DNA templates carrying a T7 promoter sequence. PCR primers used for the preparation of the DNA templates are listed in Supplementary Data file 1. sgRNAs were purified with the RNA Clean & Concentrator kit and reannealed (95 °C for 5 min, ramp to 25 °C at 0.3 °C/s). sgRNAs were quality checked using 10% denaturing polyacrylamide gels and ethidium bromide staining.

## Protein purification
All SpCas9 variants were subcloned from pMJ806 (Addgene #39312)[7] [except pET-HypaR-SpCas9-NLS-6xHis, which was subcloned in pET-Cas9-NLS-6xHis (Addgene #62933) plasmid]. For detailed cloning information and sequence information see Methods: Plasmid construction section, Supplementary Data file 1 and Supplementary Notes. The resulting fusion constructs contained an N-terminal hexahistidine (His6), a Maltose binding protein (MBP) tag and a Tobacco etch virus (TEV) protease site (except pET-HypaR-SpCas9-NLS-6xHis).

The expression constructs of the SpCas9 variants were transformed into *E. coli* BL21 Rosetta 2 (DE3) cells, grown in Luria-Bertani (LB) medium at 37 °C for 16 h. 10 ml from this culture was inoculated into 1 l of growth media (12 g/l Tripton, 24 g/l Yeast, 10 g/l NaCl,

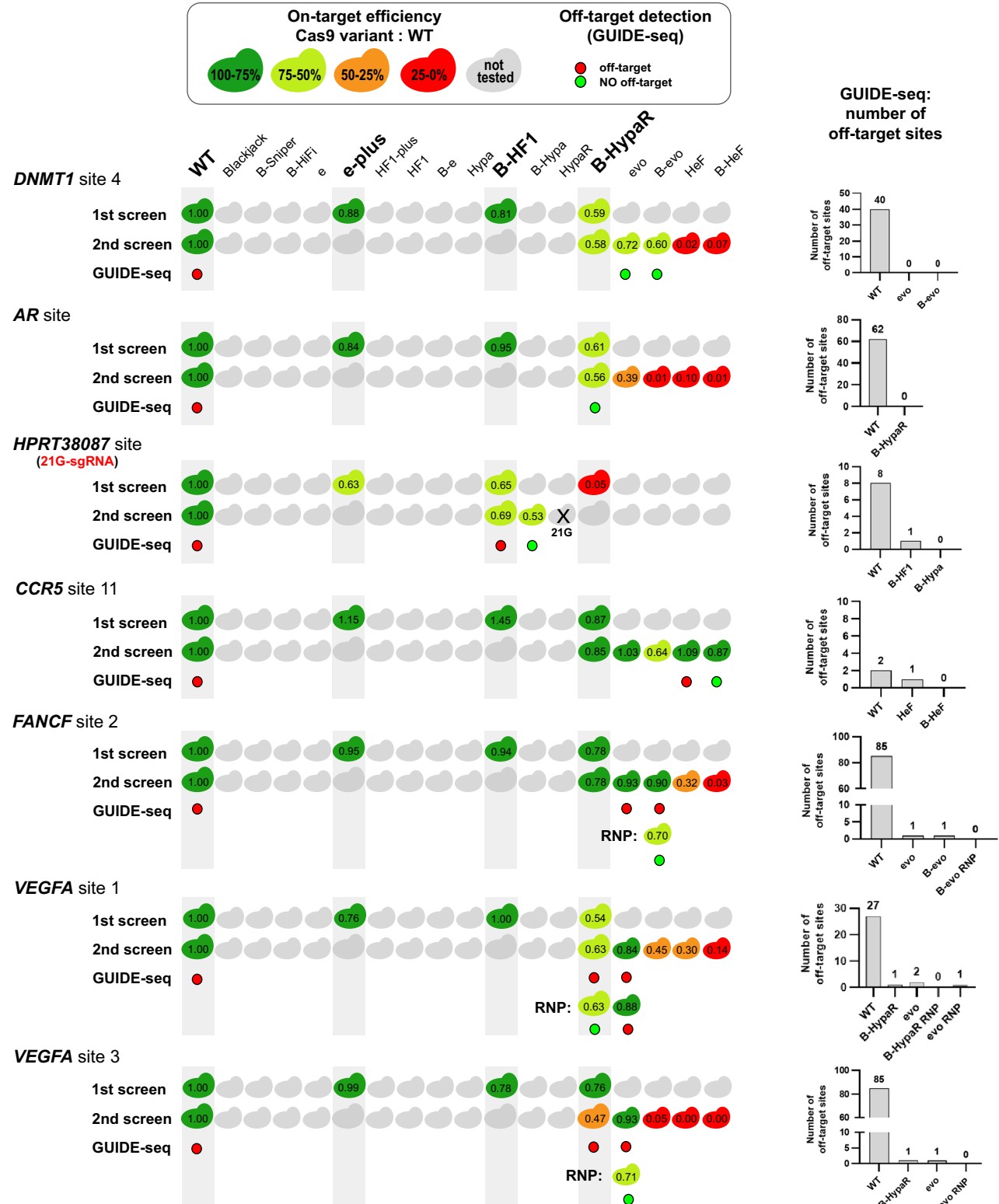

**Fig. 7 | Even repetitive, non-typical sequences can be edited without off-targets by employing optimal target-matched IFNs.** Targets shown here are a collection of targets that had previously only been edited by IFNs with off-targets detected by GUIDE-seq[20,21,23,24,26,28]. Here, they were all successfully edited without any genome-wide off-targets when assessed by GUIDE-seq using target-matched IFNs. The numbers in the colored Cas protein illustrations indicate the percentage value of the on-target genome modifications normalized to WT (measured by NGS). GUIDE-seq was performed with one or two target-matched IFNs that reached at least 50%

normalized on-target editing. Colored circles indicate whether a target was edited with (red) or without (green) off-targets in the GUIDE-seq experiment. Some targets can be edited with no detectable genome-wide off-targets by more than one target-matched IFN, or in other cases by an IFN in RNP form that can further increase specificity. Bar charts of the total number of off-target sites detected by GUIDE-seq are shown on the right side of the panel. Data related to Fig. 8, Supplementary Figs. 10, 11. Target sequences, NGS and GUIDE-seq data are reported in Supplementary Data files 1, 5, 6.

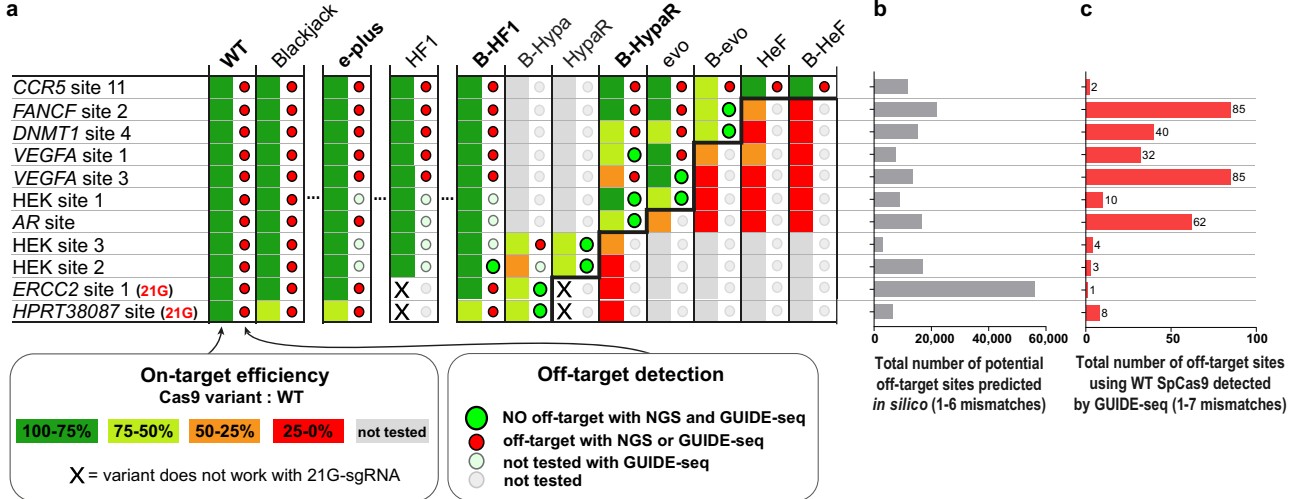

**Fig. 8 | Target-matched nucleases show high efficiency without any genome-wide off-target for targets tested in this study regardless their ranking.**
**a** Summary of targets edited by IFNs, examined in this study with GUIDE-seq and NGS. For 10 target sites, including those challenging targets where previous attempts with IFNs had failed, we were able to perform editing without any off-target detected by GUIDE-seq and further confirmed by NGS on the top three site. For the highest ranked target, *CCR5* site 11, NGS still identified residual off-target activity even with the highest ranked B-HeF, indicating that development of an even higher fidelity IFN would be beneficial for accessing the highest cleavability rank. The colors of the squares indicate the percentage value of the on-target genome

modification normalized to WT (measured by NGS). Colored circles indicate the summarized GUIDE-seq and NGS results; green circle indicates when both NGS and GUIDE-seq showed no off-targets, red circle indicates off-target editing detected either by GUIDE-seq or NGS, light green circle indicates when no off-target was found but it was only tested by NGS and gray circle indicate no data. Off-target editing data of these targets (GUIDE-seq experiments) from the literature are summarized in Supplementary Data file 6: Data from other studies. The ranking of the targets is weakly related to either **b**, the number of their predicted off-target sites, or **c**, the detected off-target sites using WT SpCas9 (for details see Supplementary Table 2). **a-c** Data are related to Figs. 6, 7, 9, Supplementary Figs. 9–11.

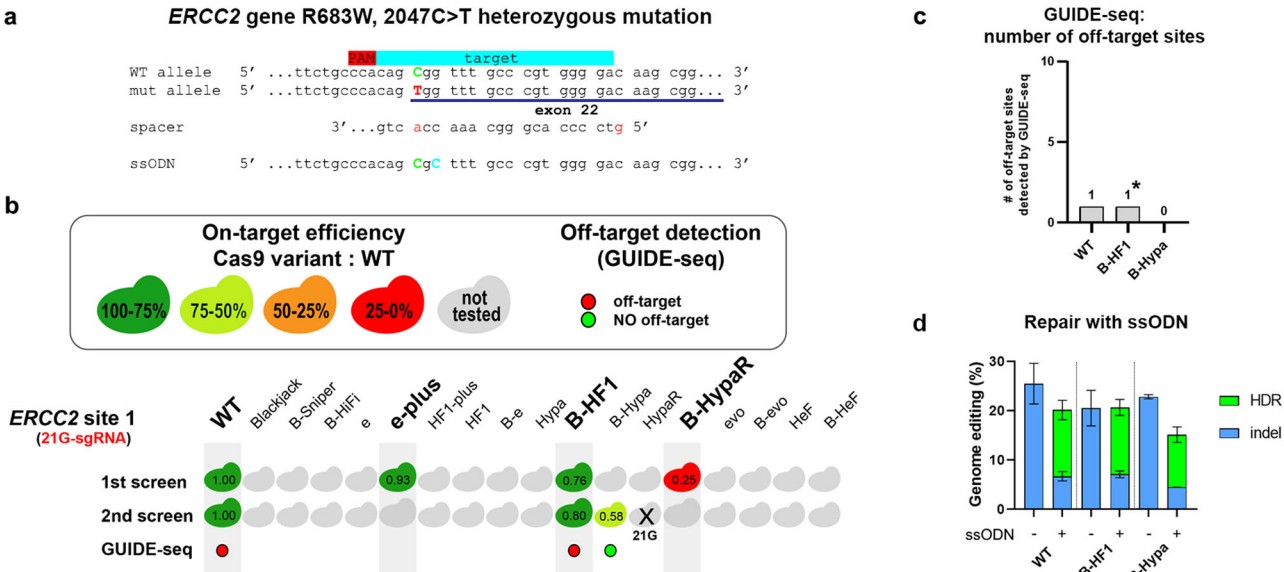

**Fig. 9 | Correcting a clinically relevant mutation without off-target cleavage using the two-step screening method. a** The strategy to correct the mutation causing Xeroderma pigmentosum in patient-derived fibroblast cells is shown, including the sequence environment of the mutation (disease-causing mutation – red letter, WT nucleotide– green letter, silent mutation – blue letter). **b** By using the two-step screening method we identified B-HypaSpCas9 to be used for editing without any detectable genome-wide off-target. Values in the colored Cas protein illustrations indicate the percentage value of the on-target genome modifications normalized to WT (measured by NGS). Hypa-R being a non-Blackjack-IFN exhibited diminished (<0.2) activity with the 21G-sgRNA (data not shown here, data available

in Supplementary Data file 5). Colored circles indicate whether a target was edited with (red) or without (green) off-targets in the GUIDE-seq experiment. **c** Bar chart showing the total number of off-target sites detected by GUIDE-seq. '*' indicates that no off-target was detected in a repeated GUIDE-seq experiment, even though the read numbers were higher in the repeated experiment (see Supplementary Fig. 11). **d** B-HypaSpCas9 with single strand oligo nucleotide repair using HDR enhancer M3814[86] provides WT-like level of correction of the R683W (2047C>T) mutation. Means and SD are shown; n = 3. **a–d** Data related to Fig. 8, Supplementary Figs. 10, 11. Target sequences, NGS and GUIDE-seq data are reported in Supplementary Data files 1, 5, 6.

**Table 1 | Different sets of SpCas9 variants (details with Addgene numbers can be found in the Materials and methods section and in Supplementary Data file 1)**

| Set A variants | Set B variants | CRISPRecise (Set C) variants | Additional in-between variants |
|---|---|---|---|
| WT SpCas9 | WT SpCas9 | WT SpCas9 | Sniper SpCas9 |
| eSpCas9-plus | Blackjack SpCas9 | Blackjack SpCas9 | HiFi SpCas9 |
| B-SpCas9-HF1 | SpCas9-HF1 | B-Sniper SpCas9 | B-evoSpCas9-V495M |
| B-HypaR-SpCas9 | B-HypaSpCas9 | B-HiFi SpCas9 | B-evoSpCas9-N515Y |
|  | evoSpCas9 | eSpCas9 | B-evoSpCas9-E526K |
|  | B-evoSpCas9 | eSpCas9-plus | B-evoSpCas9-Q661R |
|  | B-HeFSpCas9 | SpCas9-HF1-plus | B-HeFSpCas9-A661R |
|  |  | SpCas9-HF1 | B-HeFSpCas9-A695Q |
|  |  | B-eSpCas9 | B-HeFSpCas9-A848K |
|  |  | HypaSpCas9 | B-HeFSpCas9-A926Q |
|  |  | B-SpCas9-HF1 | B-HeFSpCas9-A1003K |
|  |  | B-HypaSpCas9 | B-HeFSpCas9-A1060R |
|  |  | HypaR-SpCas9 |  |
|  |  | B-HypaR-SpCas9 |  |
|  |  | evoSpCas9 |  |
|  |  | B-evoSpCas9 |  |
|  |  | HeFSpCas9 |  |
|  |  | B-HeFSpCas9 |  |

883 mg/l $NaH_2PO_4$ $H_2O$, 4.77 g/l $Na_2HPO_4$, pH 7.5) and cells were grown at 37 °C to a final cell density of 0.6 OD600, and then were cooled to 18 °C. The protein was expressed at 18 °C for 16 h following induction with 0.2 mM IPTG. Proteins were purified by a combination of chromatographic steps by NGC Scout Medium-Pressure Chromatography Systems (Bio-Rad). The bacterial cells were centrifuged at 6,000 rcf for 15 min at 4 °C. The cells were resuspended in 30 ml of Lysis Buffer (40 mM Tris pH 8.0, 500 mM NaCl, 20 mM imidazole, 1 mM TCEP) supplemented with Protease Inhibitor Cocktail (1 tablet/30 ml; complete, EDTA-free, Roche) and sonicated on ice. Lysate was cleared by centrifugation at 48,000 rcf for 40 min at 4 °C. Clarified lysate was bound to a 5 ml Mini Nuvia IMAC Ni-Charged column (Bio-Rad). The resin was washed extensively with a solution of 40 mM Tris pH 8.0, 500 mM NaCl, 20 mM imidazole, and the bound proteins were eluted by a solution of 40 mM Tris pH 8.0, 250 mM imidazole, 150 mM NaCl, 1 mM TCEP. 10% glycerol was added to the eluted sample and the His6-MBP fusion proteins were cleaved by TEV protease (3 h at 25 °C) (except pET-HypaR-SpCas9-NLS-6xHis). The volume of the protein solution was made up to 100 ml with buffer (20 mM HEPES pH 7.5, 100 mM KCl, 1 mM DTT). Proteins were purified on a 5 ml HiTrap SP HP cation exchange column (GE Healthcare) and eluted with 1 M KCl, 20 mM HEPES pH 7.5, 1 mM DTT. They were then further purified by size exclusion chromatography on a Superdex 200 10/300 GL column (GE Healthcare) in 20 mM HEPES pH 7.5, 200 mM KCl, 1 mM DTT and 10% glycerol. The eluted protein was confirmed by SDS-PAGE and Coomassie brilliant blue R-250 staining, and they were stored at −20 °C.

**Determining active SpCas9 quantity in solution**
The quantification method was based on Liu et al.[84]. The quantity of active SpCas9 protein in solution was determined using EGFP target site 32, that has shown high cleavage activity with all three proteins tested based on previous experiments. The measurement procedure is as follows: The target plasmid was incubated for an hour with protein-sgRNA complex, in different concentrations. Concentrations were determined by spectrophotometry (Nanodrop OneC), and then the target site containing the plasmid (10 nM) and the SpCas9 protein were mixed in a ratio between 1:0.5 and 1:10, while the quantity of the sgRNA

was twice that of the protein in each case. To terminate cleavage reaction, the inactivation solution (final concentration: 0.2% SDS, 50 mM EDTA) was added to the reaction mix at 80 °C. Samples were ran on a 0.8% agarose gel. Following densitometry (GelQuantNET, BiochemLabSolutions.com), the ratio of intact plasmid and total DNA was calculated for each sample. These values were plotted and fitted on a 'One-phase exponential decay function with time constant parameter' curve in Origin 2018. Taken the results of this experiment, the active SpCas9 variant quantities in solution were calculated. It was also taken into consideration that SpCas9 has a one-fold turnover rate.

**Determining cleavage rate of WT, B-HF1 and B-evoSpCas9 variants in vitro**
At first, two different solutions were made: (1) target site containing plasmid solution and (2) an SpCas9-sgRNA master mix. After mixing them (see below) the ratio of the target site containing plasmid and active protein was 1:2. Both solutions were diluted with the same cleavage buffer (final concentration: 20 mM HEPES pH 7.5, 200 mM KCl, 2 mM $MgCl_2$, 1 mM TCEP, 2% glycerol) and were pre-incubated at 37 °C before reaction. To trigger cleavage reaction, the target site containing plasmid solution was added to the SpCas9-sgRNA mixture. To terminate cleavage reaction the inactivation solution (final concentration: 0.2% SDS, 50 mM EDTA) was added to the reaction mix at 80 °C at different time points. In case of the WT SpCas9 protein the sampling points were between 2 and 30 s, while in case of the increased-fidelity SpCas9 variants they fell between 5 s and 2 h. To determine sampling points precisely a digital chronometer was attached to the pipette which can record time points in an application developed by us. This precise time determination was only necessary in the case of WT SpCas9 due to the fast reaction rate. Samples were then ran on a 0.8% agarose gel. Following densitometry (GelQuantNET, BiochemLabSolutions.com), the ratio of intact plasmid and total DNA was calculated for each sample. These values were plotted and fitted on a 'One-phase exponential decay function with time constant parameter' curve in Origin 2018. Experiments were performed in triplicates. All fitted curves are available in Supplementary Fig. 6, the k values are available in Supplementary Data file 8.

## Cell culturing and transfection

Cells employed in the studies are HEK293 (Gibco 293-H cells), GM08207 (Coriell Cell Repositories, Simian virus 40-transformed XP-D fibroblast), N2a.dd-EGFP (a neuro-2a mouse neuroblastoma cell line developed by us containing a single integrated copy of an EGFP-DHFR[DD] [EGFP-folA dihydrofolate reductase destabilization domain] fusion protein coding cassette originating from a donor plasmid with 1000 bp long homology arms to the *Prnp* gene driven by the *Prnp* promoter (*Prnp*.HA-EGFP-DHFR[DD]), N2a.EGFP and HEK-293.EGFP (both cell lines containing a single integrated copy of an EGFP cassette driven by the *Prnp* promoter)[33] cells. Cells were grown at 37 °C in a humidified atmosphere of 5% $CO_2$ in high glucose Dulbecco's Modified Eagle medium (DMEM) supplemented with 10% heat inactivated fetal bovine serum, 4 mM ʟ-glutamine (Gibco), 100 U/ml penicillin and 100 µg/ml streptomycin. Cells were passaged up to 20 times (washed with PBS, detached from the plate with 0.05% Trypsin-EDTA and replated). After 20 passages, cells were discarded. Cell lines were not authenticated as they were obtained directly from a certified repository or cloned from those cell lines. Cells were tested for mycoplasma contamination.

Cells were plated in case of each cell line one day prior to transfection in 48-well plates at a density of approximately $2.5–3 × 10^4$ cells/well. Cells were co-transfected with two types of plasmids: SpCas9 variant expression plasmid (137 ng) and sgRNA and mCherry coding plasmid (97 ng) using 1 µl TurboFect reagent according to the manufacturer's protocol. For negative control experiments either deadSpCas9 plasmid was co-transfected with a targeting sgRNA plasmid, or active SpCas9 variant with a non-targeting sgRNA plasmid. Transfection efficacy was calculated via mCherry expressing cells. Transfections were performed in triplicates. Transfected cells were analyzed ~96 h post transfection by flow cytometry and genomic DNA was purified according to the Puregene DNA Purification protocol (Gentra systems).

## Plasmid and ribonucleoprotein electroporation

Briefly, $2 × 10^5$ cells were resuspended in transfection solution (see below) and mixed with 666 ng of SpCas9 variant expression plasmid and 334 ng of sgRNA and mCherry coding plasmid. In the case of GUIDE-seq experiments an additional 30 pmol dsODN (according to the original GUIDE-seq protocol[20]) was added to the mixture. For negative control experiments either a deadSpCas9 plasmid was co-transfected with a targeting sgRNA plasmid, or an active SpCas9 variant with a non-targeting sgRNA plasmid. Nucleofections were performed in the case of HEK293, GM08207 and HEK-293.EGFP cell lines using the CM-130 program on a Lonza 4-D Nucleofector instrument on strip, either with 20 µl SF solution according to the manufacturer's protocol, or with 20 µl homemade nucleofection solution as described in Vriend et al.[85]. Transfection efficacy was calculated via mCherry expression. Unless noted otherwise, transfected cells were analyzed ~96 h post transfection by flow cytometry followed by genomic DNA purification according to the Puregene DNA Purification protocol (Gentra systems) and downstream applications such as on-target amplicon PCR in three technical replicates.

In the case of EGFP 43, *FANCF* site 2 and *VEGFA* site 2 WT SpCas9 and SpCas9-HF1 GUIDE-seq experiments the electroporation was done as follows. Briefly, $2 × 10^6$ HEK293.EGFP or HEK293 cells were resuspended with 3 µg of SpCas9 variant expressing plasmid, 1.5 µg of mCherry and sgRNA coding plasmid and 100 pmol of the dsODN mixed together with 100 µl homemade nucleofection solution as described in Vriend et al.[85]. The mixture was electroporated using Nucleofector 2b (Lonza) with A23 program and 2 mm electroporation cuvettes.

*VEGFA* site 2 B-evo dRNA experiments were based on Rose et al.[75]. *VEGFA* sgRNA2 OT1 dRNA3 was used as follows: $1 × 10^6$ HEK293 cells were resuspended in 100 µl SF solution and mixed with 2.5 µg of

B-evoSpCas9 expression plasmid and in case of dRNA 1:1 ratio: 1250 ng of dRNA3 and mCherry coding plasmid and 1250 ng of *VEGFA* site 2 sgRNA and mCherry coding plasmid, and in case of dRNA 6:1 ratio: 3000 ng of dRNA3 and mCherry coding plasmid and 500 ng of *VEGFA* site 2 sgRNA and mCherry coding plasmid. An additional 150 pmol GUIDE-seq dsODN was added to the mixture. Nucleofections were performed using the CM-130 program on a Lonza 4-D Nucleofector instrument in cuvettes according to the manufacturer's protocol. Transfected cells were analyzed ~48 h post-transfection by flow cytometry. EGFP 43 WT, e- and SpCas9-HF1, *FANCF* site 2 WT, e-plus and HF1-plus and *VEGFA* site 2 WT and SpCas9-HF1 experiments are also described in Kulcsár et al.[21].

In the case of RNP experiments with *VEGFA* site 1 B-HypaR- and evoSpCas9 RNP, *VEGFA* site 2 B-evo SpCas9 RNP and *VEGFA* site 3 evoSpCas9 RNP, $2 × 10^5$ HEK293 cells were transfected with 40 pmol SpCas9 and 48 pmol sgRNA (*VEGFA* site 1 and 3 in conditions with RNP 20 pmol SpCas9 and 24 pmol sgRNA), which was complexed in Cas9 storage buffer (20 mM HEPES pH 7.5, 200 mM KCl, 1 mM DTT and 10% glycerol) for 15 min at RT. 30 pmol of the dsODN was mixed with 20 µl SF solution to the RNP complex and electroporated using the CM-130 program on a Lonza 4-D Nucleofector instrument on strip. In case of *VEGFA* site 2 B-evo SpCas9 RNP, transfected cells were analyzed ~24 h post-transfection by flow cytometry. In the case of RNP experiments with EGFP 43 B-evo SpCas9 RNP, *FANCF* site 2 B-evo SpCas9 RNP, $2 × 10^6$ HEK293 or HEK293.EGFP cells were transfected with 100 pmol SpCas9 and 120 pmol sgRNA, which was complexed in Cas9 storage buffer (20 mM HEPES pH 7.5, 200 mM KCl, 1 mM DTT and 10% glycerol) for 15 min at RT. 100 pmol of the dsODN was mixed together with 100 µl homemade nucleofection solution to the RNP complex and electroporated using Nucleofector 2b (Lonza) with A23 program and 2 mm electroporation cuvettes.

## Flow cytometry

Flow cytometry analyses were carried out on an Attune NxT Acoustic Focusing Cytometer (Applied Biosystems). For data analysis Attune NxT Software v.2.7.0 was used. Single cells were gated based on side and forward light-scatter parameters and a total of 5000 to 10,000 viable single cell events were acquired in all experiments. The GFP fluorescence signal was detected using the 488 nm diode laser for excitation and the 530/30 nm filter for emission, the mCherry fluorescent signal was detected using the 488 nm diode laser for excitation and a 640LP filter for emission or using the 561 nm diode laser for excitation and a 620/15 nm filter for emission. For detailed flow cytometry gating information see Supplementary Fig. 1.

## EGFP disruption assay

EGFP disruption experiments were conducted in N2a.EGFP cells for the on-target screen (see details below), and in N2a.dd-EGFP cells for the mismatch screen with. Data of the EGFP disruption experiments are available in Supplementary Data file 2, processed data of EGFP disruption experiments are available in Supplementary Data file 3, heatmap data are available in Supplementary Data file 4.

Background EGFP loss was determined for each experiment using co-transfection of dead SpCas9 expression plasmid and different targeting sgRNA and mCherry coding plasmids. EGFP disruption values were calculated as follows: the average EGFP background loss from dead SpCas9 control transfections made in the same experiment was subtracted from each individual treatment in that experiment and the mean values and the standard deviation (SD) were calculated from them. Results were normalized to the WT SpCas9 data from the same experiment.

On-target activity was measured in *N2a.EGFP* cell line. Cells were co-transfected with two types of plasmids: SpCas9 variant expression plasmid (137 ng) and sgRNA and mCherry coding plasmid (97 ng) using 1 µl TurboFect reagent per well in 48-well plates. Transfected cells were

analyzed ~96 h post-transfection by flow cytometry. In this cell line the EGFP disruption level is not saturated, this way this assay is a more sensitive reporter of the intrinsic activities of these nucleases compared to N2a.dd-EGFP cell line.

In the case of mismatch screens *N2a.dd-EGFP* cells were co-transfected with two types of plasmids: with SpCas9 variant expression plasmid (137 ng) and a mix of 3 sgRNAs in which one nucleotide position was mismatched to the target using all 3 possible bases and mCherry coding plasmid (3 × ~33.3 ng = 97 ng) using 1 μl TurboFect reagent per well in 48-well plates. TMP (trimethoprim; 1 μM final concentration) was added to the media ~48 h before FACS analysis. Transfected cells were analyzed ~96 h post-transfection by flow cytometry. Some of the data have also been shown in Kulcsár et al.[21]. The 4-day post-transfection results with this cell line show a close to saturated level, this way it is a good reporter system for seeing the full spectrum of off-target activities.

### Processing data from the study of Kim et al.

Data from Kim et al.[34] in Fig. 3c–g were processed as follows. In case of the on-target screen, we selected those targets that were interrogated with perfectly matching tRNA-N$_{20}$ protospacers (6481 target sites) to avoid 5′ mismatched sgRNAs, then we excluded those targets that either lack data for any of the nucleases or were cleaved by the WT SpCas9 with lower than 15% indel occurrence.

In case of the mismatch screen, we processed the data as follows. We calculated the average of the on-target modification rates normalized to the corresponding WT values from the parallel experiments, and for further processing, we selected data from only those off-targets and IFNs, where the corresponding average on-target values normalized to the WT were at least 0.20 measured on day 4. We considered only the one base mismatching targets: off-targets with every possible one base mismatch for all positions, i.e., 60 data points per sgRNA, and for all the 30 sgRNAs per SpCas9 variant (i.e., 1800 datapoints overall). The average of the modification (indel) percentages of the 60 off-target values for each sgRNA and IFN pair were calculated and normalized to the corresponding on-target value of the SpCas9 variants on day 7. These are presented along with the day 7 on-target data in the heatmap in Fig. 3f. For detailed information see Supplementary Data file 7.

### Bioinformatic tool development for the prediction of target ranking

For prediction, a long short-term memory (LSTM) network was used to perform multiclass classification. For training, outliers were removed from the data that have been selected from the DeepCRISPR database, as described above, for Fig. 3c. The model was trained on the training set (5466) and tested on the test set (948) as separated in Kim et al.[34]. The bases were coded as one-hot labels and classes were created based on the number of proteins that cut the sequence. During training, the number of epochs were determined by early stopping.

### On-target heatmaps

The algorithms for ordering rows and columns on the on-target heatmaps is the following: After subtracting the background, normalized on-target values were calculated by dividing them with the WT value and then rounding them to two decimals. Values that were below zero were rounded to zero. Values lower than 0.20 were regarded as no cleavage. Heatmaps were ordered as follows: (i) IFNs were ordered according to how many targets they could cleave. When the number of cleaved targets was the same for multiple IFNs, they were ordered according to their average normalized on-target activity. (ii) Targets were ordered based on the number of IFNs that can cleave them, taking it into consideration to minimize the number of outliers. On each heatmap, a bold line shows the threshold between cleaved and non-cleaved datapoints, and outliers are clearly indicated.

### Binary classification

G-mean is the squared root of the product of the sensitivity and specificity that was calculated for the entire on-target heatmap for the cleaved and non-cleaved groups, where the bold line indicates where the cleavability law predicts the border between cleaved (≥0.20) and non-cleaved (<0.20) values. For G-mean calculation data (confusion matrix, sensitivity and specificity) see Supplementary Data file 4.

ROC curves are graphs that plot a model's false-positive rate against its true-positive rate across a range of classification thresholds. ROC curves were generated for individual columns of the on-target heatmaps representing the normalized on-target activity values for a variant to assess how accurately the cleavage rule ordered its targets into cleaved and non-cleaved classes. For ROC curve and AUC calculation data see Supplementary Data file 4.

### ssODN repair of *ERCC2* exon22 R683W (2047C>T) mutation

Donor ssODN for GM08207 cell line *ERCC2* exon22 R683W (2047C>T) mutation repair was designed to have the wild type base and a silent mutation (to identify the repair outcome). The 90 nt long ssODN was centered at the desired mutations (Fig. 9a, d and Supplementary Data file 1: PCR primers/ERCC2 90 nt + marked primer). Briefly, 2 × 10$^5$ GM08207 cells were resuspended in 20 μl homemade nucleofection solution as described in Vriend et al.[85] and mixed with 666 ng of SpCas9 variant expression plasmid and 334 ng of sgRNA and mCherry coding plasmid and 2 μl of 100 μM ssODN donor. Nucleofections were performed using the CM-130 program on a Lonza 4-D Nucleofector instrument. Cells were plated in 48-well plates containing 0.5 ml of completed DMEM and 2 μM M3814 HDR enhancer[86] (which was a kind gift from Stephan Riesenberg) per well. After 2 days media was changed to fresh completed DMEM. Transfections were performed in triplicates. For negative control experiments deadSpCas9 plasmid was co-transfected with the targeting sgRNA plasmid. Transfected cells were analyzed ~96 h post-transfection by flow cytometry and genomic DNA was purified according to the Puregene DNA Purification protocol (Gentra systems). For NGS data information see Supplementary Data file 5 and NGS sequencing data are deposited at NCBI Sequence Read Archive: PRJNA1008914.

### Indel analysis by next-generation sequencing (NGS)

Amplicons for deep sequencing were generated using two rounds of PCR to attach Illumina handles. The 1st step PCR primers used to amplify target genomic sequences are listed in Supplementary Data file 1: PCR primers. PCR was done in a S1000 Thermal Cycler (Bio-Rad) or PCRmax Alpha AC2 Thermal Cycler using the by Q5 high-fidelity polymerase with supplied Q5 buffer (in case of VEGFA site 2 amplicon together with Q5 High GC enhancer) and 150 ng of genomic DNA in a total volume of 25 μl. The thermal cycling profile of the PCR was: 98 °C 30 s; 35 × (denaturation: 98 °C 20 s; annealing: see Supplementary Data file 1: PCR primer, 30 s; elongation: 72 °C, see Supplementary Data file 1: PCR primer); 72 °C 5 min. i5 and i7 Illumina adapters were added in a second PCR reaction using Q5 high-fidelity polymerase with supplied Q5 buffer (in case of VEGFA site 2 amplicon together with Q5 High GC enhancer) and 1 μl of first step PCR product in total volume of 25 μl. The thermal cycling profile of the PCR was: 98 °C 30 s; 35 × (98 °C 20 s, 67 °C 30 s, 72 °C 20 s); 72 °C 5 min. Amplicons were purified by agarose gel electrophoresis. Samples were quantified with Qubit dsDNA HS Assay kit and pooled. Double-indexed libraries were sequenced on a MiSeq, MiniSeq or NextSeq (Illumina) giving paired-end sequences of 2 × 150 bp or 2 × 250 bp, it was performed by ATGandCo or Deltabio Ltd. Reads were aligned to the reference sequence using BBMap. Indels were counted computationally amongst the aligned reads that matched at least 75% of the first 20 bp of the reference amplicon. Indels without mismatches were searched starting at ±2 bp around the cut site. For each sample, the indel frequency was determined as (number of reads with an indel)/(number of total reads). The 15 bp long center

fragment of the GUIDE-seq dsODN sequence ("gttgtcatatgttaa"/"ttaa-catatgacaac") was counted in the aligned reads to measure dsODN on-target tag integration for GUIDE-seq experiments. The ssDNA repair was determined as (number of reads with desired edit)/(number of total reads). Results can be found in Supplementary Data file 2. The following software were used: BBMap 38.08, samtools 1.8, BioPython 1.71, PySam 0.13. For NGS data information see Supplementary Data file 5 and NGS sequencing data are deposited at NCBI Sequence Read Archive: PRJNA1008914.

## GUIDE-seq

GUIDE-seq relies on the integration of a short dsODN tag into DNA breaks, therefore after the genomic DNA purification, dsODN tag integration and efficient indel formation was verified in the on-target site by NGS. In the next step genomic DNA was sheared with BioraptorPlus (Diagenode) to 550 bp in average. Sample libraries were assembled as previously described[20] and sequenced on Illumina MiSeq or MiniSeq instrument by ATGandCo or Deltabio Ltd. Data were analyzed using open-source guideseq software (version 1.1)[87]. Consolidated reads were mapped to the human reference genome GrCh37 supplemented with the integrated EGFP sequence. Upon identification of the genomic regions integrating double-stranded oligodeoxynucleotide (dsODNs) in aligned data, off-target sites were retained if at most seven mismatches against the target were present and if absent in the background controls. Visualization of aligned off-target sites are provided as a color-coded sequence grid. Summarized results can be found in Supplementary Data file 6 and GUIDE-seq sequencing data are deposited at NCBI Sequence Read Archive: PRJNA1008914.

## Statistics

Differences between SpCas9 variants were tested by using either two-tailed paired-samples Student's t-test (Fig. 5b SpCas9-NG/SpCas9-NG-L847R-V1015R) or by using two-tailed Wilcoxon Signed Ranks test (Fig. 5b HypaR/HypaR-L847R-V1015R, xCas9/xCas9-L847R-V1015R) in the cases where differences did not meet the assumptions of Paired t-test. Differences between groups were tested by using either two-tailed unpaired Student's t-test with Welch's correction (Fig. 4c B-SpCas9-HF1) or by using two-tailed Mann–Whitney test (Fig. 4c B-evoSpCas9) in the cases where differences did not meet the assumptions of unpaired t-test. Differences between SpCas9 variants were tested by using RM one-way ANOVA and Dunnett's multiple comparisons test with a single pooled variance (Fig. 3b) or by using RM one-way ANOVA, with the Geisser-Greenhouse correction and Dunnett's multiple comparisons test with individual variances computed for each comparison (Fig. 5a) or (ii) Tukey's multiple comparisons test with individual variances computed for each comparison (where the mean of each column was compared with the mean of every other columns: Figs. 2b, 3d, Supplementary Figs. 2i, 10) in the cases where sphericity did not meet the assumptions of RM one-way ANOVA. Differences between more than two groups were tested by using Kruskal-Wallis test (Figs. 2e, 3g, Supplementary Fig. 3d). Normality of data and of differences was tested by Shapiro–Wilk normality test. Statistical tests were performed using GraphPad Prism 9 on data including all parallel sample points. Test results are shown in Supplementary Data file 9.

## Reporting summary

Further information on research design is available in the Nature Portfolio Reporting Summary linked to this article.

## Data availability

Expression vectors developed in this study are available from Addgene: Expression plasmids for human codon-optimized increased-fidelity SpCas9 variants: B-Sniper SpCas9 (#207361), B-HiFi SpCas9 (#207362) HypaR-SpCas9 (Addgene #126757), B-HypaR-SpCas9 (Addgene #126764, B-evoSpCas9-V495M (#207363), B-evoSpCas9-

N515Y (#207364), B-evoSpCas9- E526K (#207365), B-evoSpCas9-Q661R (#207366), B-HeFSpCas9-A661R (#207367), B-HeFSpCas9-A695Q (#207368), B-HeFSpCas9-A848K (#207369), B-HeFSpCas9-A926Q (#207370), B-HeFSpCas9-A1003K (#207371), B-HeFSpCas9-A1060R (#207372). Expression of increased-fidelity SpCas9 variants in bacterial cells: WT SpCas9 (#207373), Sniper SpCas9 (#207374), Blackjack SpCas9 (#207375), HiFi SpCas9 (#207376), B-Sniper SpCas9 (#207377), B-HiFi SpCas9 (#207378), eSpCas9 (#207379), eSpCas9-plus (#207380), SpCas9-HF1-plus (#207381), SpCas9-HF1 (#207382), B-eSpCas9 (#207383), HypaSpCas9 (#207384), B-SpCas9-HF1 (#207385), B-HypaSpCas9 (#207386), HypaR-SpCas9 (#207387), B-HypaR-SpCas9 (#207388), evoSpCas9 (#207389), B-evoSpCas9 (#207390), HeFSpCas9 (#207391), B-HeFSpCas9 (#207392). The CRISPRecise set (Set C; see Table 1), which contains the IFN set proposed here to facilitate efficient editing of practically all target sites with no off-target effects detectable by GUIDE-seq in these research setups, is available from Addgene as the CRISPRecise plasmid kit. The deep sequencing data are available in NCBI Sequence Read Archive: PRJNA1008914. Source Data are provided in the Supplementary Data files and Source Data file. Source data are provided with this paper.

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

## Acknowledgements

We thank Ildikó Szűcsné Pulinka, Judit Szűcs, Vivien Karl, Lilla Burkus, Barbara Karsai, Judit Kálmán for their excellent laboratory assistance, Dorottya Simon, Antal Nyeste, Edit Szabó, György Várady, Diána Szeregnyei, Katalin Reith for their valuable help. We thank Stephan Riesenberg for his valuable advice and providing the HDR enhancer[86]. We thank Dóra Bokor for proofreading the manuscript. We thank Viktória Faragó for her valuable help in figure design. This research was supported by grants K128188, K134968, and K142322 to E.W and PD134858 to P.I.K. from the Hungarian Scientific Research Fund (OTKA) and P.I.K. by 2018-1.1.1-MKI-2018-00167 and by ÚNKP-20-5-SE-20 from the National Research, Development and Innovation Office. P. I. K. is a recipient of the János Bolyai Research Scholarship of the Hungarian Academy of Sciences (BO/764/20). S.L.K was supported by grant EFOP-3.6.3-VEKOP-16-2017-00009 from the Higher Education Institutional Excellence Program of the Semmelweis University.

## Author contributions

P.I.K. and E.W. conceived and designed experiments, interpreted the results, P.I.K., A.T., Z.L., E.T., R.Z., Z.B., V.L.V., S.L.K., K.H. performed all experiments. P.I.K., A.T., E.T., Z.L., R.Z., S.L.K analyzed the data. A.W. performed the bioinformatic tool development. P.I.K. and E.W. wrote the manuscript with input from all the authors.

## Funding

## Competing interests

The authors declare no competing interests.
