## [Peer Review File · Nature Communications]

Reviewers' Comments:

Reviewer #1:

Remarks to the Author:

The authors describe a set of data comparing on- and off-target cleavage of SpCas9 and various engineered variants thereof. From this data, the authors create a series of rankings of the various enzymes and introduce a "cleavage rule." The authors propose using similar search strategies to guide researcher choice of the most effective Cas9 variants to balance efficacy and specificity.

Overall, while the work presents a large data set, the utility and impact of the study is unclear in light of the current literature. More rigorous methodology could be applied, in particular in statistical comparisons of the data sets and in the measurement of off-target cutting in other ways besides GUIDE-seq. The impact to researcher gene editing methodology is not apparent.

Major comments

Experiments were performed via plasmid transfection. However, results can differ when the Cas9 protein is delivered in other means, such as via RNP, in which the protein dose is effectively lower. It would be beneficial to confirm the rank order efficacy of a subset of gRNA/protein pairs as measured via RNP delivery. There is some apparent use of RNP later in the paper (Figure 9), but the included data is confusing and difficult to compare. The authors argue that they have comparison data for dose in an additional preprint; however this data is likewise with plasmid delivery and the kinetics of long-term Cas9 expression off of the plasmid may be distinct from a shorter burst. This data also does not appear to show dose effect on off-targets, which may plateau at differing doses between the variants.

The author's arguments would be bolstered by a scatter plot correlation and associated correlation values of on-target efficacy versus specificity for the variants.

The language around a "cleavage rule" is confusing: is there novel findings here? It seems that the authors are simply suggesting that cleavage is a product of an inherent enzymatic activity and target sequence. I would suggest removing this language and the associated figure 1.

The authors touch deeply on topics found otherwise, particularly for example in the cited article by Schmid-Burgk et al 2020. This article describes activity/specificity comparisons in depth, as do some others. However, the findings are not obviously addressed in the text.

Again, comparing the GUIDE-seq data and GFP disruption would be better served by including scatter plot comparisons and associated statistical correlations. Likewise the comparison data from Kim et al might be better compared by Spearman rank correlation.

Biochemical studies in Figure 5 would likewise better be served by dose-curve studies rather than kinetics due to the typical lack of turnover by the Cas9 protein.

In comparing the PAM variants, are outlying data points overemphasized by the colorbar scheme as combined with the lower starting efficacy as normalized to WT cas9? Are outliers significant? Again, a more objective correlative statistical treatment would be preferred to that provided by the heat map color.

The authors' claims that GUIDE-seq has superior off-target finding as compared to methods such as Digenome should be tempered down: Digenome and other in vitro methods have been shown to find off-targets that GUIDE-seq can miss. This does not invalidate the authors' decision to use GUIDE-seq due to the high number of combinations.

Although the GUIDE-seq data is encouraging for indicating no off-targets above detection, amplicon sequencing (perhaps at the top 3 off-targets) would be much more sensitive in parallel to support the claim that off-target editing is sub-detection. The read count for GUIDE-seq reads is occasionally sparse, especially for the ERCC2 gene described in Figure 10 (and this should be

acknowledged or Figure 10C should be replaced with read counts). Since the focus of the paper is on these fidelity measurements, an orthologous measurement is strongly encouraged.

Generally, the discussion could be greatly condensed.

The application the authors suggest is unclear: "selecting the IFN that is closest to a target-matched one." How does this study inform this behavior in practice? It seems the suggestion is that researchers still screen many variants and choose the most appropriate one with sufficient activity. This seems to already be the state of the art: how does this data impact that pipeline?

Minor comments

The two color scales are confusing: the "beneficial" side of the arrow would be better aligned (better efficiency and better specificity to the right, for example).

For experiments challenging off-target cleavage as in Supplemental Figure 3C, are the spacers delivered in pool or averaged from arrayed testing? The wording appears ambiguous.

Axis labels are very difficult to read (Ex: Supplemental Figure 2, Figure 2, etc)

Reviewer #2:

Remarks to the Author:

Since the discovery of the CRISPR-Cas9 gene editing technology, how to increase on-target activity and decrease its off-target effect have been the two major focuses of improvements centered the CRISPR gene editing society. Hence, a number of increased fidelity nucleases (IFNs) based on the WT SpCas9 have been developed. Most of these IFNs in their original study were claimed to retain similar on-target activity as the WT Cas9 while significantly improve specificity. In this study, the authors have performed an excellent comparison and analysis of these IFNs and reported the finding that the increased fidelity SpCas9 nucleases (IFNs) follow an on-target and off-target cleavage ability rule.

Using a EGFP disruption assay, the authors quantified the on-target efficiency of 50 sgRNAs and the off-target activities of 162 sgRNAs with one mismatch for seven IFNs. Based on the finding of interconnected on-target and off-target activities of the IFNs, the authors increased the resolution of target-matched IFNs by generating a new set of Cas9 variants (19 IFNs). To validate the target-dependent on-target and off-target characteristics of the IFNs, the study performed genome-wide off-target screening using GUIDE-seq, tested the on-target activities on 52 endogenous target sites, as well as performed independent analysis of a large dataset of on-target and off-target activities generated by Kim et al. 2020. Ten new IFN variants were further created by the authors to further increase the resolution of IFNs in the lower/medium fidelity range. Based on the total of 29 IFN variants, the authors demonstrated that by employing a target-matched strategy, of which on-target activities was firstly assessed by with the four categories of Cas9 IFNs followed by neighboring IFN on-target activity assay. The IFNs with high fidelity potential while still exhibiting reasonably high on-target activities were selected for genome-wide off-target assay with GUIDE-seq. The study demonstrated that using this target-matched approach IFNs with no detectable genome-wide off target effect can be identified for a given sgRNA, including sgRNAs which had been previously shown challenging reducing off-targets by IFNs. Lastly, the study demonstrated with an example that they can identified an optimal target-matched IFN to achieve efficient correction of point mutation in the ERCC2 gene without detectable genome-wide off-target effect.

In general, this is a very plausible work including a substantial number of original experiments and novel findings. The new IFNs generated by the study are valuable for the big CRISPR gene editing society. Thus, it is important that these newly generated IFNs are made available in public plasmid depositary e.g., Addgene. It is convincing that the IFNs can be categorized into four groups based on their on-target cleavage ability and levels of fidelity depending the on the targeting sgRNAs. However, it might not be practical and will be quite laborious for gene editing experiments that

two-rounds of on-target activities must be performed for a number of these IFNs. As already discussed by the authors, it will be more useful and practical if predictive algorithm is available for identification of target-matched IFN variants. For a given sgRNA, one can predict which IFN will be the optimal target-matched one with no off-target and good on-target activity. Below, I highlighted a few comments to the authors for the further improvement of their manuscript.

1. I am a little bit concerned about the over-stated conclusion of “any target” without off-target effect in the title and the abstract. This might be challenged by the number of target sites evaluated, ways of delivering and expressing the IFNs, and methods for the genome-wide OT detection and evaluations. For example, site 2 from the Kim dataset still exhibit a high degree of off-target effect with the evo variant.

2. In assay of the on-target activities of the 50 EGFP sgRNAs and the 6481 sgRNAs from the Kim dataset, is it possible to identify a certain sequence pattern or predictive rule that categorize the gRNAs into cleavable or non-cleavable from the highly specific IFN variants.

For example, in Figure 4c, there are already more than 3000 on-target data points available for developing a prediction model for evo IFN. For sgRNA with predicted activity above the on-target activity threshold, IFNs within the evo fidelity ranking should be selected. For sgRNAs predicted with low and no on-target efficiency for the evo IFNs, then a prediction is performed for the next-level of reduced fidelity IFNs (Hypa groups). Since this is a binary prediction model, it should be quite straightforward to explore such a predictive model. Alternatively, the authors can analyze the prediction score of these sgRNAs from deepCRISPRvariants developed by Kim et al. in their study.

3. The study has used GUIDE-seq for genome-wide screening of off-targets. It seems that the sequencing depth for the different IFN experiments are not equal. For example, in Figure S5 and S9, total reads of the WT are much higher than other IFN variant groups. This might have an impact on the off-target detection.

4. It is not clear to me that why the study used the MS2 version of the sgRNA. The MS2 scaffold is developed for the SAM CRISPRon system. Any reason of why using the MS2 gRNA scaffold instead of the normal SpCas9 gRNA scaffold?

5. It is also not clear if all the IFN variants are expressed with the same promoter and the poly A signal. In the results, authors concluded that there is not different in the expression levels of the IFNs in the cells by referring to Supplementary Figure 4. However, the results in Supplementary Figure 4 do not really support this conclusion. For example, some of IFN variants (B-Snipper, B-Hypa) are clearly lower than WT SpCas9. It is important to quantify and make sure that these IFNs and WT are expressed at the same level in the tested cells, as it will have a significant impact on both on-target and off-target activity.

6. Figure 3A should be correct to make bars align with the text in the X axel. Also, plotting out other IFN variants which were not tested by GUIDE-seq in this part makes the figure confusing. Alternatively, those IFNs not tested here can change to gray text font.

7. There is a mistake in Figure 5A. Supercoil plasmid should be the smaller and intense band, while the upper one is nicked plasmid.

8. Other small comments: (a) The resolution of figures is generally low, which makes it difficult to evaluate the details. (b) The figures should be better ordered. E.g., Heatmap scalebar legend for activity and specificity in Figure 2 and 4 were placed above the figure. Figure 5, the way of ordering the panel a, b, c looks very stochastic. (c) Color code for on-target efficiency range in Figure 7, 8, 10 is not easy. Consider using range instead, for example, 0-20%, 20-50%, 50-70%, >70%.

Reviewer #3:

Remarks to the Author:

In their manuscript, Kulcsar and colleagues describe a method to identify Cas9 variants (out of a set of 19, including several newly described variants) that have a balanced sense of on- and off-target activities. In concept, this is a really nice study, and it would be great for the field to have a predictable way to tune on- and off-target activity. However, this manuscript, at least as presented, does not appear to describe such a method. While a "cleavage rule" is motivated and an experimental algorithm is described (Figure 7a), the cleavage rule comes with many caveats (five are presented by the authors themselves in the discussion), and the algorithm is not followed in the manuscript and also comes with caveats (several presented in the discussion). Therefore, it is difficult to see the value of the authors' cleavage rule.

Major comments:

1) The title of this manuscript makes a broad claim - that the authors are describing a method to "edit any target without genome-wide off-target effects." This manuscript does not show this, and the authors need to revise the title for this manuscript to be publishable

In the authors' discussion about the use of GUIDE-seq to quantify off-targets, they state that the sensitivity of GUIDE-seq is "compatible to or even higher than that of amplicon sequencing" and cite reference 59. This is a misreading of the reference, which clearly states "When more sensitive validation methods are used, it is clear that the complementary biochemical methods can detect some bona fide sites of off-target activity that are not picked up by GUIDE-seq. These types of methods are recommended when the highest sensitivity is required or in cell types that are sensitive to being transfected with dsODN tags."

GUIDE-seq has higher specificity for bona fide off-target loci, but it is well known that it is less sensitive than Digenome-seq, CIRCLE-seq, and CHANGE-seq, and therefore GUIDE-seq alone cannot be used to make the statement that there are no "genome-wide off-target effects." Frankly, even those methods paired with locus-by-locus validation cannot support the statement "no genome-wide off-target effects", since that is a question of analytical sensitivity due to cellular input and sequencing error.

2) In terms of real-world application (Figure 7), there is a disconnect between what is written in the text (lines 312-323) and what is shown in Figure 7b.

2a) In step 1, the user should screen with e-plus, B-Hypa, B-HypaR, according to the authors. This is done in Figure 7b. Following this, one is supposed to "identify which one has the highest fidelity, that still shows sufficient efficiency." Then one is supposed to "test on-target activity of "a few additional IFNs, neighbouring the one selected in the first step"

For HEK site 1, all four screening nucleases had $\geq 85\%$ efficiency, so the authors presumably took the next four on the list. This should be explained in the text why they went with the next four rather than flanking four.

For HEK site 2, the choice of e-plus seems reasonable, as does the choice to take all five nucleases between e-plus and B-Hypa, though this is a lot of screening to do. There is a drop off from 105% to 25% in the first screen.

For HEK site 3, the choice of the 2nd screen does not make sense. E-plus was 92% and B-Hypa was 67%. The reasonable choice would have been to test the same five enzymes that were tested for HEK site 2. The authors need to explain this choice, since otherwise the data seem cherry-picked. For example, despite the cleavage rule, B-Hypa is at 67% and HypaR is at 95%, while the next in line (B-HypaR) is at 32%. I would not have screened HypaR, based on what is written in the text, but the authors end up choosing this as the optimal enzyme.

For VEGFA site 1, the 2nd screen choice also does not make sense. I would have chosen the five enzymes between wt and e-plus. And this is also a case where the cleavage rule has failed, since the on-target values for the five sites chosen (in order) are 69%, 98%, 126%, 86%, 78%, and these are bracketed by 73% and 51% from the first screen.

2ai) The data for HEK site 1 and VEGFA site 1 suggest that the authors may have chosen the

wrong four enzymes as the first screen, or at least need to include one from the highest fidelity group (Sniper, Blackjack, HiFi, B-Sniper, B-Hifi). If the cleavage rule works well, I would expect one of those would have a higher efficiency than the 87%, 91%, and 85% (or 73%, 51%, 53%) from e-plus, B-Hypa, B-HypaR.

2b) The final step is "out of these variants, the optimal target-matched variant with the maximum specificity is selected and/or confirmed by GUIDE-seq." There is no comment in the main text about how the authors chose "optimal target-matched variant."

For HEK site 1, the authors chose B-HypaR, which had the highest rate (85%) of the 2nd screen nucleases, and it had no GUIDE-seq off-targets. This seems like a straightforward case.

For HEK site 2, the authors chose to test Hypa (91%) and B-HF1 (72%). I suppose this could be rationalized by the fact that these are the two least highly active of the 2nd screen, and both have no GUIDE-seq off-targets. This seems like a win.

For HEK site 3, as mentioned above, the authors make an unexpected choice of HypaR in the second screen. It has 95% activity, but would not have been a reasonable choice based on the cleavage rule, since it is ranked lower than the screening B-Hypa nuclease at 67%. A more consistent choice for GUIDE-seq would have been B-HF1, at 76%, and Hypa at 95%. It will be important for the authors to explain these decisions and to perform GUIDE-seq for HEK site 3 with Hypa and B-HF1.

For VEGFA site 1, the authors also made an unexpected choice of B-HypaR. First, as mentioned above, the cleavage rule has not worked, as there seems to be no trend in cleavage rate. That said, it may be reasonable to choose B-HF1, and then flank it with B-Hypa for the purposes of comparison for the paper. Both, however, have off-target sites by GUIDE-seq (see comment 3). Unexpectedly, the authors also tested B-HypaR, even though it should have been screened out after the 1st screen, and this enzyme had no GUIDE-seq off-targets. The authors need to clarify these decisions, since they contradict what would be expected by the method described in the manuscript. Additional GUIDE-seq data may be useful as well.

3) The authors claim that "Figure 7b demonstrates that with the panel of 19 IFNs, all four targets could be edited without any genome-wide off-target effect detected by GUIDE-seq." This is a well-parsed, factually-correct statement. But it is not what the authors describe in Figure 7a. It is not reasonable to expect a lab to do GUIDE-seq with 19 different Cas9 variants and then choose the best one. Or, if it is, labs can already do this with a large subset of the 19, without relying on this manuscript. Even if one does the full method, as described, it would involve testing 8-9 variants, and doing GUIDE-seq on 2-3 of them.

3a) The experiments in Figure 8 do not seem to follow at all the strategy described in Figure 7a, so it is hard to assess whether the authors' method works. A comment is made in the figure legend that three targets "had sufficient information to skip the rough screen". A similar comment is made in the main text. The authors don't explicitly state what that information is, but in any case, since the authors are essentially presenting the method described in Figure 7a, that exact method should be used in Figure 8.

3b) The data in Figure 8 for CCR5 site 11, FANCF site 2, and VEGFA site 3 also do not appear to follow the cleavage rule. I would expect there to be a lot more correlation than there is between adjacent nucleases (see comment 4). I also would expect that more of the targets at the border between green and blue or green and orange would have an absence of GUIDE-seq off-targets, if the cleavage rule is relevant, and this does not seem to be the case.

4) Figure 9 actually clearly shows that the authors did not follow their recommendations of using WT, e-plus, B-Hypa, and B-HypaR as the first screening set.

5) The authors seem to have done a very limited set of off-target sites to establish the cleavage rule (looking at mismatches in only the first few positions in the target site). To establish that it is going to be generalizable, the authors should extend this analysis through all positions of the

target site. This may contribute to some of the cleavage rules failures that are seen in Figures 7 and 8.

6) I agree that off-target activity is a concern, but I think language like "raises potential safety concerns" is more accurate than "poses safety concerns" (lines 40-41), since the functional consequences of off-target editing are yet to be understood. Similarly, the phrase "pose a major limitation" (line 17) may be overstated, as clearly clinical trials are being allowed to move forward, and not all are using a high-fidelity Cas9.

We are grateful to the three reviewers for the time they devoted to improving our manuscript and appreciate the help their comments provided.

Reviewer #1 (Remarks to the Author):

The authors describe a set of data comparing on- and off-target cleavage of SpCas9 and various engineered variants thereof. From this data, the authors create a series of rankings of the various enzymes and introduce a 'cleavage rule'. The authors propose using similar search strategies to guide researcher choice of the most effective Cas9 variants to balance efficacy and specificity.

Overall, while the work presents a large data set, the utility and impact of the study is unclear in light of the current literature. More rigorous methodology could be applied, in particular in statistical comparisons of the data sets and in the measurement of off-target cutting in other ways besides GUIDE-seq. The impact to researcher gene editing methodology is not apparent.

Major comments

Experiments were performed via plasmid transfection. However, results can differ when the Cas9 protein is delivered in other means, such as via RNP, in which the protein dose is effectively lower. It would be beneficial to confirm the rank order efficacy of a subset of gRNA/protein pairs as measured via RNP delivery. There is some apparent use of RNP later in the paper (Figure 9), but the included data is confusing and difficult to compare. The authors argue that they have comparison data for dose in an additional preprint; however this data is likewise with plasmid delivery and the kinetics of long-term Cas9 expression off of the plasmid may be distinct from a shorter burst. This data also does not appear to show dose effect on off-targets, which may plateau at differing doses between the variants.

Response: In our opinion, the two-dimensional ranking pattern shown in Figure 2 cannot be caused by a dose effect or a plateau effect. Nevertheless, we agree with the reviewer that the experiments presented do not show the extent to which these factors might contribute to the ranking pattern shown in the figures. Therefore, in the revised version we show that the different activities of the variants are not the result of dose dependence (new Supplementary Fig. 7; page 12, line 257) by using 4-5 targets with different variants, including both cutting and non-cutting ones. Furthermore, we also agreed with the reviewer, in that "to confirm the rank order efficacy of a subset of gRNA/protein pairs as measured via RNP delivery" would be beneficial. Therefore, we tested the WT and 9 variants in pre-assembled RNP form on 11 targets in an on-target disruption screen (Supplementary Fig. 8a, b; page 13, line 276), on 8 targets in a mismatch screen with three mismatching guides per target (Supplementary Fig. 8c, d) and on 2 targets and their most significant off-target by amplicon sequencing (Supplementary Fig. 8e, f). Interestingly, in the case of some targets (in addition to what are shown in the figures) RNP activity was so low even with WT SpCas9, that it prevented reliable interpretation of the results. We checked the activity of the sgRNAs again *in vitro*, repeated the experiments, but we consistently got low values with these targets. The targets, where the WT disruption value was less than 25%, were omitted from the figures but are listed in Supplementary Table 3 and 5.

As expected, lower EGFP disruption % (Supplementary Fig. 8a, b) as well as lower modification % (Supplementary Fig. 8e, f) were obtained with RNP than with the slower-decaying transient plasmid transfection, however, the order of the variants and the targets remained the same with RNP (Supplementary Fig. 8). Out of the 99 points tested, the RNP disruption experiment resulted in one single outlier (RNP G-mean score of 0.98, plasmid G-mean score of 1.00; Supplementary Fig. 8a, b).

Using EGFP disruption assay for 8 targets and 24 single mismatched sgRNAs, we also demonstrated that the cleavage rule remains evident with variants in pre-assembled RNP form (Supplementary Fig. 8c, d).

These results are in line with our expectation that, although besides the three main factors (fidelity-increasing mutations, target sequence contribution and mismatches in the case of off-targets), several other factors may modulate IFN activity, the effect of the main factors, the cleavage rule, remain clearly discernible.

The figures are also shown here.

Supplementary Figure 7

Supplementary Figure 7. IFN-coding plasmid titration

Titration of plasmid amounts expressing wild-type SpCas9 and different IFN variants on selected targets on which the IFN has either WT-like activity or no/low activity. Means are shown for each data point, error bars represent the standard deviation (SD) for triplicates. Target sequences, raw and processed disruption data are reported in Supplementary Table 1-3.

Supplementary Figure 8

Supplementary Figure 8. The cleavability rule is also apparent and an identical order of targets and IFNs is observed with the RNP as with the plasmid form when tested with 9 IFNs.

a-b, Heatmaps show the normalized EGFP disruption activities of SpCas9 nucleases with perfectly matching 20G-sgRNAs in case of RNP (a) or plasmid (b) form. The bold line indicates the dividing line

defined by the cleavage rule between the classes of cleaved and not-cleaved values. **c-d**, Heatmaps show the normalized EGFP disruption activities of SpCas9 nucleases either with perfectly matching (blue to red) or with one-base mismatching (yellow to brown) 20G-sgRNAs in case of RNP (**c**) or plasmid (**d**) form. Mismatch screen of the nuclease variants either with perfectly matching 20G-sgRNAs or with mismatched sgRNAs (a mixture of three different sgRNAs used for each examined mismatch position as indicated in the figure). Grey boxes: not determined because on-target activity was too low. **e-f**, Bar chart of the results of on-target and the top off-target site detected by NGS in case of WT and nine IFNs alongside with various negative controls. Means and SD are shown; n=3 (overlaid as white circles). Target and primer sequences, raw and processed disruption and NGS data are reported in Supplementary Tables 1-5.

The author's arguments would be bolstered by a scatter plot correlation and associated correlation values of on-target efficacy versus specificity for the variants.

Response: Since we presented the fidelity ranking of the variants and the trade-off between activity and specificity at the Keystone Symposia in Banff in February 2020, a few studies have confirmed this either on larger on-target datasets (Kim et al., 2020; Kim et al., 2023) or on genome-wide off-targets of a larger number of targets (Schmid-Burgk et al., 2020). Here, however, we demonstrated the activity and specificity of IFNs as a function of the targets. At the two extremes are the highly cleavable targets, that can be cleaved by all IFNs but only the highest fidelity ones cleave them with high specificity, and the hardly cleavable targets, that can only be cleaved by the lowest fidelity IFNs but without many off-targets. In our opinion, a heatmap is a better way to convey this message than a scatter plot.

Kim, N., Kim, H.K., Lee, S., Seo, J.H., Choi, J.W., Park, J., Min, S., Yoon, S., Cho, S.R., and Kim, H.H. (2020). Prediction of the sequence-specific cleavage activity of Cas9 variants. *Nat Biotechnol* 38, 1328-1336.
Kim, Y.H., Kim, N., Okafor, I., Choi, S., Min, S., Lee, J., Bae, S.M., Choi, K., Choi, J., Harihar, V., et al. (2023). Sniper2L is a high-fidelity Cas9 variant with high activity. *Nat Chem Biol*.
Schmid-Burgk, J.L., Gao, L., Li, D., Gardner, Z., Strecker, J., Lash, B., and Zhang, F. (2020). Highly Parallel Profiling of Cas9 Variant Specificity. *Mol Cell* 78, 794-800 e798.

The language around a 'cleavage rule'; is confusing: is there novel findings here? It seems that the authors are simply suggesting that cleavage is a product of an inherent enzymatic activity and target sequence. I would suggest removing this language and the associated figure 1.

Response: Yes, there are new findings here, sorry if we were not able to explain them clearly. We now have tried to clarify this in the revised MS. Firstly, we would like to point out that the cleavage rule does not seem to work for PAM mutants, although it can also be said that cleavage is simply a product of an inherent enzymatic activity and target sequence.

Our results show that the effects of mismatches and fidelity-increasing mutations, as well as the contribution of the target sequence, are additive, allowing this two-dimensional pattern to emerge. This implies that these three major factors have similar mechanisms of action. The fact that mismatches and fidelity-increasing mutations in eSpCas9 and SpCas9-HF1 arrest the nuclease in an inactive intermediate state and slow down the formation of a cleavage-competent conformation by the movement of the HNH domain has been shown previously, e.g., Dagdas et al., 2017; Chen et al., 2017. It is also known that the formation of the perfectly complementary spacer-target

DNA hybrid helix triggers this transition. Our results suggest that (i) mutations in the other IFNs studied here also exert their effects by a similar mechanism (independent of the nature of the mutations); and that (ii) target sequence contributions also act by inducing this transition from an inactive to an active intermediate state. Thus, different target sequences may activate the nuclease by inducing the docking of the HNH domain to the cleavage-competent intermediate state to different extents.

The difference in the activating effect of the target sequences is not pronounced for WT SpCas9, in which the HNH domain is stably docked in the active conformation even with smaller sequence contributions. However, in the variants with lower activity than WT, where the transition from an inactive to an active state is slower, the different activating effects of the target sequences are clearly manifested, as our data show.

This does not necessarily mean that "cleavage is simply a product of inherent enzymatic activity and target sequence". For example, SpCas9 activity is also influenced by self-complementary or scaffold complementary motifs in spacer sequences (Huszar et al., 2023; Thyme et al., 2016). These motifs either block sgRNA loading or DNA binding, thus affecting the activity of the nuclease by an apparently different mechanism. Several additional factors can affect IFN cleavage, a significant proportion of which also affect the activity of the WT SpCas9 and which can only be partially minimized by normalizing the activity of the variants to the activity of the WT SpCas9. In addition, amino acid mutations in the variants may form specific interactions with particular target/spacer sequences or with specific mismatches, the effect of which is independent of the more general sequence contributions found here. Several studies have argued for such effects of mutations in the variants (Bravo et al., 2022; Kim et al., 2023; Lee et al., 2018). This effect can sometimes be large enough to cause the appearance of some outliers. These additional factors are all included in the definition that "cleavage is a product of the inherent enzyme activity and the target sequence".

In contrast, we show that although such specific effects exist, the general activating effect of the sequences is the dominant force in the vast majority of cases. This is a novel finding with important implications.

Chen, J.S., Dagdas, Y.S., Kleinstiver, B.P., Welch, M.M., Sousa, A.A., Harrington, L.B., Sternberg, S.H., Joung, J.K., Yildiz, A., and Doudna, J.A. (2017). Enhanced proofreading governs CRISPR-Cas9 targeting accuracy. *Nature* 550, 407-410.

Dagdas, Y.S., Chen, J.S., Sternberg, S.H., Doudna, J.A., and Yildiz, A. (2017). A conformational checkpoint between DNA binding and cleavage by CRISPR-Cas9. *Sci Adv* 3, eaao0027.

Gong, S., Yu, H.H., Johnson, K.A., and Taylor, D.W. (2018). DNA Unwinding Is the Primary Determinant of CRISPR-Cas9 Activity. *Cell Rep* 22, 359-371.

Huszar, K., Welker, Z., Gyorgypal, Z., Toth, E., Ligeti, Z., Kulcsar, P.I., Dancso, J., Talas, A., Krausz, S.L., Varga, E., et al. (2023). Position-dependent sequence motif preferences of SpCas9 are largely determined by scaffold-complementary spacer motifs. *Nucleic Acids Res.*

Thyme, S.B., Akhmetova, L., Montague, T.G., Valen, E., and Schier, A.F. (2016). Internal guide RNA interactions interfere with Cas9-mediated cleavage. *Nat Commun* 7, 11750.

Bravo, J.P.K., Liu, M.S., Hibshman, G.N., Dangerfield, T.L., Jung, K., McCool, R.S., Johnson, K.A., and Taylor, D.W. (2022). Structural basis for mismatch surveillance by CRISPR-Cas9. *Nature* 603, 343-347.

Kim, Y.H., Kim, N., Okafor, I., Choi, S., Min, S., Lee, J., Bae, S.M., Choi, K., Choi, J., Harihar, V., et al. (2023). Sniper2L is a high-fidelity Cas9 variant with high activity. *Nat Chem Biol.*

Lee, J.K., Jeong, E., Lee, J., Jung, M., Shin, E., Kim, Y.H., Lee, K., Jung, I., Kim, D., Kim, S., et al. (2018). Directed evolution of CRISPR-Cas9 to increase its specificity. *Nat Commun* 9, 3048.

The authors touch deeply on topics found otherwise, particularly for example in the cited article by Schmid-Burgk et al 2020. This article describes activity/specificity comparisons in depth, as do some others. However, the findings are not obviously addressed in the text.

Response: As discussed above, three in-depth studies have been published that partially support our findings presented at the Keystone conference (Kim et al., 2020; Kim et al., 2023; Schmid-Burgk et al., 2020). The study by Schmid-Burgk et al. was indeed the first article to meticulously show an activity-fidelity relationship using 8 increased-fidelity SpCas9 variants including xCas9, and also showed that this PAM mutant variant has a slightly different activity-fidelity relationship than the other seven increased-fidelity variants. Their genome-wide off-target detection methods allowed them to assess the specificity of IFNs on a much larger set of targets than in other studies. We described this and cited their studies on page 4, line 72 in the revised MS.

Kim, N., Kim, H.K., Lee, S., Seo, J.H., Choi, J.W., Park, J., Min, S., Yoon, S., Cho, S.R., and Kim, H.H. (2020). Prediction of the sequence-specific cleavage activity of Cas9 variants. *Nat Biotechnol* 38, 1328-1336.

Kim, Y.H., Kim, N., Okafor, I., Choi, S., Min, S., Lee, J., Bae, S.M., Choi, K., Choi, J., Harihar, V., et al. (2023). Sniper2L is a high-fidelity Cas9 variant with high activity. *Nat Chem Biol*.

Schmid-Burgk, J.L., Gao, L., Li, D., Gardner, Z., Strecker, J., Lash, B., and Zhang, F. (2020). Highly Parallel Profiling of Cas9 Variant Specificity. *Mol Cell* 78, 794-800 e798.

Again, comparing the GUIDE-seq data and GFP disruption would be better served by including scatter plot comparisons and associated statistical correlations. Likewise the comparison data from Kim et al might be better compared by Spearman rank correlation.

Response: We respectfully disagree with the reviewer on this point. We have tried to find the representation that best fits the nature of the data. Spearman's rank correlation is sensitive to the small spread of WT-like activities, which may obscure the difference between working and non-working activities.

Biochemical studies in Figure 5 would likewise better be served by dose-curve studies rather than kinetics due to the typical lack of turnover by the Cas9 protein.

Response: Due to the typical lack of turnover of the Cas9 protein, the points of a dose curve only show the amount of active protein presented in the reaction (Gong et al., 2018), without showing much of its actual target-dependent activity. Dose-curve studies are useful for active site titration. We used the knowledge about the amount of active protein in the preparation to determine the actual ratio of active RNP:target in the kinetic studies.

Gong, S., Yu, H.H., Johnson, K.A., and Taylor, D.W. (2018). DNA Unwinding Is the Primary Determinant of CRISPR-Cas9 Activity. *Cell Rep* 22, 359-371.

In comparing the PAM variants, are outlying data points overemphasized by the colorbar scheme as combined with the lower starting efficacy as normalized to WT cas9? Are outliers significant? Again, a more objective correlative statistical treatment would be preferred to that provided by the heat map color.

Response: They are not overemphasized. There are several variants with even lower overall activity that have no outliers. Of the other 19 variants, 11 have no outliers and another 7 have only one. In contrast, these PAM mutants

have 10 outliers (out of 42 points) from both the working and non-working classes. In fact, to avoid any overemphasizing and to decrease the number of the outliers, some of the targets, amongst which the order was not strictly determined by the 19 IFNs, were reordered (compared to Fig. 2a heatmap) to favor the accommodation of SpCas9-NG and xCas9 into the heatmap, as we indicated in the Figure 5d legend in the original MS.

We use an objective statistical treatment that best fits the nature of the data: we characterize whether a variant obeys the cleavage rule by receiver operating characteristic area under the curve (AUC) values (Fig. 6e in the original MS, Fig. 5e in the revised MS).

The EGFP disruption % values of the outliers of the PAM-altered mutants are significantly different from the disruption % values of the WT and of the higher activity (and lower fidelity) neighboring variants (in the case of false negative outliers) and from the disruption % values of the lower activity neighboring variants (in the case of false positive outliers) (only 8 out of 255 datapoints were not significantly different: EGFP site 13 - xCas9 vs. e, Hypa, HypaR and B-HypaR; EGFP site 36 - SpCas9-NG vs. B-e, Hypa; EGFP site 47 - xCas9 vs. HypaR and B-HypaR).

The heatmap color is only there to help visualize the cleavage rule: In the two-dimensional pattern each column represents the cleavage of a variant on the target set in a ranked order. Each variant is able to cleave the targets (going from the top towards the bottom of the heatmap) until the sequence contribution of the targets decreases to the point where they cannot be cleaved. From this point on, no target with lower cleavability is cleaved in the given column. Similarly, each row represents the cleavage of a target by the set of variants in a ranked order. Each target is cleaved by the variants (from left to right) until the effect of fidelity-increasing mutations in a variant becomes so large that the activity of the nuclease is reduced to the point where it cannot cleave that target. After this point, no variants in that row with higher fidelity will cleave the target. This information is clearly visible from the heatmaps.

The authors claims that GUIDE-seq has superior off-target finding as compared to methods such as Digenome should be tempered down: Digenome and other *in vitro* methods have been shown to find off-targets that GUIDE-seq can miss. This does not invalidate the authors decision to use GUIDE-seq due to the high number of combinations.

Response: We agree with the reviewer on this point too. However, we may also point out here, that GUIDE-seq and *in vitro* methods, such as Digenome-seq, CIRCLE-seq typically differ in the validated low-read count off-targets with WT SpCas9 (Tsai, S. Q. et al., 2017). The target-matched, increased-fidelity variant that ultimately succeeds in editing without off-targets reduces the cleavage of even the most persistent off-targets to zero, these are ones that typically show high-read counts with both GUIDE-seq and *in vitro* methods with the WT SpCas9. Thus, the off-targets with low-read count with WT that may be seen in *in vitro* methods but not in GUIDE-seq have low relevance for our experiments. Furthermore, the sensitivity of the biochemical methods in our case is ultimately identical to amplicon sequencing, since the off-targets found in Digenome-seq and other biochemical methods need to be validated by amplicon sequencing.

Additionally, in the revised version, we performed NGS on the top three off-target sites found by GUIDE-seq, which thus, plausibly provides the same sensitivity as the biochemical methods.

Tsai, S. Q. et al. CIRCLE-seq: a highly sensitive *in vitro* screen for genome-wide CRISPR-Cas9 nuclease off-targets. *Nat Methods* 14, 607-614, doi:10.1038/nmeth.4278 (2017).

Although the GUIDE-seq data is encouraging for indicating no off-targets above detection, amplicon sequencing (perhaps at the top 3 off-targets) would be much more sensitive in parallel to support the claim that off-target editing is sub-detection.

Response: We agree with the reviewer that amplicon sequencing could support our claims, and therefore we performed the experiments they suggested on the top three off-targets. NGS confirmed the GUIDE-seq results for all targets where GUIDE-seq showed no off-target by an IFN, except for B-HeFSpCas9 with the CCR5 site 11 target, where the read count of a residual off-target was significantly greater than that of the four different negative controls ($p=0.0209, 0.0346, 0.0163, 0.0269$). CCR5 site 11 is the highest ranking target out of the 113 target we examined in this study, suggesting that not many targets belong to this ranking range. This result suggests that the development of a new variant with higher fidelity than these IFNs, higher than that of B-HeFSpCas9 may be necessary for these rare cases.

The read count for GUIDE-seq reads is occasionally sparse, especially for the ERCC2 gene described in Figure 10 (and this should be acknowledged or Figure 10C should be replaced with read counts).

Response: We repeated the GUIDE-seq experiments in the occasional cases where the read counts were lower, including the experiment for the ERCC2 gene.

The results:

Target site	SpCas9 variant	New GUIDE-seq on-target read	Old GUIDE-seq on-target read
HEK site 1	B-HypaR-SpCas9	3588	1265
HEK site 2	B-SpCas9-HF1	4946	3131
HEK site 2	HypaR-SpCas9	8533	4946
HEK site 3	B-HypaSpCas9	2941	2001
HEK site 3	HypaR-SpCas9	4368	1592
VEGFA site 1	WT SpCas9	3829	1192
VEGFA site 1	B-HypaR-SpCas9	1778	1058
VEGFA site 3	evoSpCas9	2890	1623
ERCC2 site 1	WT SpCas9	5981	291
ERCC2 site 1	B-SpCas9-HF1	2592	449

As this shown in Supplementary Figure 9a, the average on-target GUIDE-seq read count in this study is the highest among the studies examining a number of targets including the one that developed GUIDE-seq.

Since the focus of the paper is on these fidelity measurements, an orthologous measurement is strongly encouraged.

Response: We agree with the reviewer, we also did amplicon sequencing at the top off-targets as suggested above. The results are shown in Supplementary Figure 10 and summarized in Figure 8.

Generally, the discussion could be greatly condensed.

Response: We agree with the reviewer and have edited some parts of the discussion, omitted some of our points and tried to make the rest more concise.

The application the authors suggest is unclear: 'selecting the IFN that is closest to a target-matched one'. How does this study inform this behavior in practice? It seems the suggestion is that researchers still screen many

variants and choose the most appropriate one with sufficient activity. This seems to already be the state of the art: how does this data impact that pipeline?

Response: The current state of the art is to test a few IFNs, including, at most, eSpCas9, SpCas9-HF1, HypaSpCas9, evoSpCas9 and maybe Sniper and HiFi. The screening of only these variants is not sufficient to provide an optimal variant for targets from all ranges of cleavability ranks, we corroborate this claim through examples (e.g. *VEGFA* site 2, *FANCF* site 2 and *DNMT1* site 4 in Figure 8 and Supplementary Figure 10). Here, we provided additional SpCas9 variants that fill in the gaps in the fidelity ranking of IFNs. Furthermore, while it may be feasible to test all variants when considering only the above four or five IFNs, testing about 20 variants for every target is definitely not feasible. It is for this reason that we proposed a method, building on the knowledge we have accumulated for target ranking in addition to the efficiency and specificity of IFNs in this study. To minimize the number of necessary IFN testing, we omitted two IFNs from the lower fidelity region where the differences amongst IFNs were minimal and we refined our first screen.

A significant increase in specificity can be achieved by using only the 1st, rough screen, which includes 3 IFNs. Although testing both rounds of screens, which may involve 5-7 IFNs, is not a small number, the target-matching IFNs can be found with our approach without having to test all 17 variants.

Minor comments

The two color scales are confusing: the 'beneficial' side of the arrow would be better aligned (better efficiency and better specificity to the right, for example).

Response: We corrected these as the reviewer suggested.

For experiments challenging off-target cleavage as in Supplemental Figure 3C, are the spacers delivered in pool or averaged from arrayed testing? The wording appears ambiguous.

Response: In pool. We apologize for the confusion and corrected this on page 7, line 130 and in the Figure legends.

Axis labels are very difficult to read (Ex: Supplemental Figure 2, Figure 2, etc)

Response: The resolution of the figure was drastically reduced during pdf conversion. We corrected these mistakes.

We thank Reviewer #1 for their comments, that helped to improve our manuscript.

Reviewer #2 (Remarks to the Author):

Since the discovery of the CRISPR-Cas9 gene editing technology, how to increase on-target activity and decrease its off-target effect have been the two major focuses of improvements centered the CRISPR gene editing society. Hence, a number of increased fidelity nucleases (IFNs) based on the WT SpCas9 have been developed. Most of these IFNs in their original study were claimed to retain similar on-target activity as the WT Cas9 while significantly improve specificity. In this study, the authors have performed an excellent comparison and analysis of these IFNs and reported the finding that the increased fidelity SpCas9 nucleases (IFNs) follow an on-target and off-target cleavage ability rule.

Using a EGFP disruption assay, the authors quantified the on-target efficiency of 50 sgRNAs and the off-target activities of 162 sgRNAs with one mismatch for seven IFNs. Based on the finding of interconnected on-target and off-target activities of the IFNs, the authors increased the resolution of target-matched IFNs by generating a new set of Cas9 variants (19 IFNs). To validate the target-dependent on-target and off-target characteristics of the IFNs, the study performed genome-wide off-target screening using GUIDE-seq, tested the on-target activities on 52 endogenous target sites, as well as performed independent analysis of a large dataset of on-target and off-target activities generated by Kim et al. 2020. Ten new IFN variants were further created by the authors to further increase the resolution of IFNs in the lower/medium fidelity range. Based on the total of 29 IFN variants, the authors demonstrated that by employing a target-matched strategy, of which on-target activities was firstly assessed by with the four categories of Cas9 IFNs followed by neighboring IFN on-target activity assay. The IFNs with high fidelity potential while still exhibiting reasonably high on-target activities were selected for genome-wide off-target assay with GUIDE-seq. The study demonstrated that using this target-matched approach IFNs with no detectable genome-wide off target effect can be identified for a given sgRNA, including sgRNAs which had been previously shown challenging reducing off-targets by IFNs. Lastly, the study demonstrated with an example that they can identified an optimal target-matched IFN to achieve efficient correction of point mutation in the ERCC2 gene without detectable genome-wide off-target effect.

In general, this is a very plausible work including a substantial number of original experiments and novel findings. The new IFNs generated by the study are valuable for the big CRISPR gene editing society. Thus, it is important that these newly generated IFNs are made available in public plasmid depositary e.g., Addgene. It is convincing that the IFNs can be categorized into four groups based on their on-target cleavage ability and levels of fidelity depending the on the targeting sgRNAs. However, it might not be practical and will be quite laborious for gene editing experiments that two-rounds of on-target activities must be performed for a number of these IFNs. As already discussed by the authors, it will be more useful and practical if predictive algorithm is available for identification of target-matched IFN variants. For a given sgRNA, one can predict which IFN will be the optimal target-matched one with no off-target and good on-target activity. Below, I highlighted a few comments to the authors for the further improvement of their manuscript.

1. I am a little bit concerned about the over-stated conclusion of 'any target' without off-target effect in the title and the abstract. This might be challenged by the number of target sites evaluated, ways of delivering and expressing the IFNs, and methods for the genome-wide OT detection and evaluations. For example, site 2 from the Kim dataset still exhibit a high degree of off-target effect with the evo variant.

Response: Indeed, the site 2 target of the Kim dataset can only be edited with off-targets, even when edited with evoSpCas9, the highest fidelity IFN they used. If there are not enough variants used, representing each fidelity rank, then not all targets can be edited without off-targets. This is exactly what Figures 3g (former Fig. 4g) and 2e are trying to point out. Figure 3g shows that if we only use the 5 variants (on the Kim dataset), then even when using the highest fidelity working variant for each target, we can still see substantial off-target effect in some cases, although, no doubt, they are greatly reduced compared to when using the other lower fidelity variants. That is why we had to develop additional IFNs with intermediate fidelity for this study. Figure 2e shows, that if an appropriately expanded set of IFNs are used for editing, it results with significantly less off-target than a set of only 7 IFNs (Supplementary Fig. 2d). We tried to make this clearer in the revised version.

We fully agree with the reviewer, that editing without off-target effects depends on the sensitivity of detection. Thus, by looking at it with another method or by screening more cells, one might find a target that can still be edited with off-targets only. Here, we would like to mention the great paper by Klenstriver et al. entitled "High-fidelity CRISPR-Cas9 nucleases with no detectable genome-wide off-target effects", in which the authors themselves showed in the paper that the claim is in fact not completely true, not for all targets (Kleinstiver et al., 2016). Additionally, they tested on fewer targets than we did, with similar limitations. Nevertheless, readers understood this, and the title accurately indicated what a milestone this work represents in increasing the specificity of gene editing. With all due modesty, we believe that our work also represents a very significant step forward in the realization of off-target gene editing, although our claim is obviously also limited; however, readers will understand this in the context of ongoing scientific work, in the circumstances of a typical genome modification experiment. We cannot, of course, examine every single target and determine the target matched IFN that will provide editing without off-target. However, we demonstrated the principle that makes it reasonable to believe that, in practice, off-target-free editing can be achieved for the vast majority of targets of any cleavability rank with the set of IFNs we have developed. Furthermore, by following this path, additional missing-fidelity variants can be created if there is no optimal variant for some targets in the existing set, as we found exactly this for the *CCR5* site 11. Although we demonstrated this on a relatively small number of targets, these are the targets that have not yet been successfully edited without genome-wide off-target by anyone, even though several studies have attempted to do so.

The title has been modified accordingly, and we respectfully request the acceptance of the modified title. We tried to keep the core message, but with some refinements, as suggested by this reviewer, with whom we agree. It now reads as "Target-matched increased-fidelity SpCas9 variants potentiate the cleavage of any target without genome-wide off-target effects".

Kleinstiver, B.P., Pattanayak, V., Prew, M.S., Tsai, S.Q., Nguyen, N.T., Zheng, Z., and Joung, J.K. (2016). High-fidelity CRISPR-Cas9 nucleases with no detectable genome-wide off-target effects. *Nature* 529, 490-495.

2. In assay of the on-target activities of the 50 EGFP sgRNAs and the 6481 sgRNAs from the Kim dataset, is it possible to identify a certain sequence pattern or predictive rule that categorize the gRNAs into cleavable or non-cleavable from the highly specific IFN variants.

For example, in Figure 4c, there are already more than 3000 on-target data points available for developing a prediction model for evo IFN. For sgRNA with predicted activity above the on-target activity threshold, IFNs within

the evo fidelity ranking should be selected. For sgRNAs predicted with low and no on-target efficiency for the evo IFNs, then a prediction is performed for the next-level of reduced fidelity IFNs (Hypa groups). Since this is a binary prediction model, it should be quite straightforward to explore such a predictive model. Alternatively, the authors can analyze the prediction score of these sgRNAs from DeepCRISPR developed by Kim et al. in their study.

Response: We tried both recommended approaches to test the predictions on the 50 GFP targets (Table 1). Unfortunately, the predictions did not work very well, for example, for evoSpCas9, out of 32 non-cleavable targets, only 14 were correctly predicted. Similar results were obtained when we tried to use the rationale proposed by the reviewer and another deep learning algorithm (DeepRank; Table 1 and Supp. Table 7: EGFP prediction, page 21 line 502). We believe that for a more accurate prediction and to obtain specific motifs, we would need to be able to train on a significantly larger number of targets and more variants would be needed to obtain a more reliable result. We are trying to generate such data, but this is beyond the scope of present work.

Table 1. Binary classifiers of two predictions (DeepSpCas9 and the one developed in this study, DeepRank) of on-target cleavage activity of 5 IFNs on 49 targets from Figure 2a. For calculation details see Supplementary Table 7: EGFP prediction and Methods: EGFP prediction.

SpCas9 variants	G-mean		Specificity		Sensitivity	
	DeepSpCas9	DeepRank	DeepSpCas9	DeepRank	DeepSpCas9	DeepRank
Sniper SpCas9	0	0	0	0	1	1
eSpCas9	0	0	0	0	1	1
SpCas9-HF1	0,63	0,45	0,4	0,2	0,98	1
HypaSpCas9	0,71	0,41	0,5	0,17	1	1
evoSpCas9	0,64	0,71	0,44	0,5	0,94	1

3. The study has used GUIDE-seq for genome-wide screening of off-targets. It seems that the sequencing depth for the different IFN experiments are not equal. For example, in Figure S5 and S9, total reads of the WT are much higher than other IFN variant groups. This might have an impact on the off-target detection.

Response: For GUIDE-seq experiments the suggested on-target read count for maximal sensitivity is 500 (Malinin et al., 2021). We aimed to reach at least 2,000 on-target reads for our samples. Our figure shows that we worked with higher on-target read values than most of the similar studies (mean GUIDE-seq on-target reads 4,467 vs. between 922-4,009, respectively, {all targets except VEGFA site 2 data} Supplementary Fig. 9a). However, in several cases, the number of WT reads indeed was higher than for one of the variants. We repeated several GUIDE-seq experiments in the revised manuscript to obtain higher read counts.

The results:

Target site	SpCas9 variant	New GUIDE-seq on-target read	Old GUIDE-seq on-target read
HEK site 1	B-HypaR-SpCas9	3588	1265
HEK site 2	B-SpCas9-HF1	4946	3131
HEK site 2	HypaR-SpCas9	8533	4946
HEK site 3	B-HypaSpCas9	2941	2001
HEK site 3	HypaR-SpCas9	4368	1592
VEGFA site 1	WT SpCas9	3829	1192
VEGFA site 1	B-HypaR-SpCas9	1778	1058
VEGFA site 3	evoSpCas9	2890	1623
ERCC2 site 1	WT SpCas9	5981	291
ERCC2 site 1	B-SpCas9-HF1	2592	449

Malinin, N.L., Lee, G., Lazzarotto, C.R., Li, Y., Zheng, Z., Nguyen, N.T., Liebers, M., Topkar, V.V., Iafrate, A.J., Le, L.P., et al. (2021). Defining genome-wide CRISPR-Cas genome-editing nuclease activity with GUIDE-seq. *Nat Protoc* 16, 5592-5615.

4. It is not clear to me that why the study used the MS2 version of the sgRNA. The MS2 scaffold is developed for the SAM CRISPRon system. Any reason of why using the MS2 gRNA scaffold instead of the normal SpCas9 gRNA scaffold?

Response: We apologize for this error, which was overlooked and was due to copying and pasting that sentence from a previous study that used this methodology. This error has been corrected. Detailed information about sgRNA spacer cloning can be found in Supplementary Information.

5. It is also not clear if all the IFN variants are expressed with the same promoter and the poly A signal. In the results, authors concluded that there is not different in the expression levels of the IFNs in the cells by referring to Supplementary Figure 4. However, the results in Supplementary Figure 4 do not really support this conclusion. For example, some of IFN variants (B-Snipper, B-Hypa) are clearly lower than WT SpCas9. It is important to quantify and make sure that these IFNs and WT are expressed at the same level in the tested cells, as it will have a significant impact on both on-target and off-target activity.

Response: All IFNs were expressed from the same backbone with the same promoter and codon usage. We emphasize it now on page 12 in the revised MS. Their cDNA sequences differ only at the mutated positions. We made all reasonable effort to provide comparable expression patterns for them and rigorously monitored the transfection efficiency in each experiment for all samples. Furthermore, we made transfection with diluted plasmid amounts, which showed that the observed activity differences amongst IFNs is not due to the differences in their expression levels. We also used pre-assembled RNP form with selected variants and targets to avoid the caveat described here by the reviewer (page 13 line 276). The results are shown in Supplementary Figure 7 and 8.

Supplementary Figure 7

Supplementary Figure 7. IFN-coding plasmid titration

Titration of plasmid amounts expressing wild-type SpCas9 and different IFN variants on selected targets on which the IFN has either WT-like activity or no/low activity. Means are shown for each data point, error bars represent the standard deviation (SD) for triplicates. Target sequences, raw and processed disruption data are reported in Supplementary Table 1-3.

Supplementary Figure 8

Supplementary Figure 8. The cleavage rule is also apparent and an identical order of targets and IFNs is observed with the RNP as with the plasmid form when tested with 9 IFNs.

a-b, Heatmaps show the normalized EGFP disruption activities of SpCas9 nucleases with perfectly matching 20G-sgRNAs in case of RNP (a) or plasmid (b) form. The bold line indicates the dividing line

defined by the cleavage rule between the classes of cleaved and not-cleaved values. **c-d**, Heatmaps show the normalized EGFP disruption activities of SpCas9 nucleases either with perfectly matching (blue to red) or with one-base mismatching (yellow to brown) 20G-sgRNAs in case of RNP (**c**) or plasmid (**d**) form. Mismatch screen of the nuclease variants either with perfectly matching 20G-sgRNAs or with mismatched sgRNAs (a mixture of three different sgRNAs used for each examined mismatch position as indicated in the figure). Grey boxes: not determined because on-target activity was too low. **e-f**, Bar chart of the results of on-target and the top off-target site detected by NGS in case of WT and nine IFNs alongside with various negative controls. Means and SD are shown; n=3 (overlaid as white circles). Target and primer sequences, raw and processed disruption and NGS data are reported in Supplementary Tables 1-5.

6. Figure 3A should be correct to make bars align with the text in the X axel. Also, plotting out other IFN variants which were not tested by GUIDE-seq in this part makes the figure confusing. Alternatively, those IFNs not tested here can change to gray text font.

Response: We changed the figure to make it clearer and simpler (now Supplementary Fig. 4).

Supplementary Figure 4

Supplementary Figure 4. The mismatch tolerance of IFNs seen in GFP disruption well approximates their genome-wide off-target effects.

a-c, Four targets and various IFNs were selected from the on-target and mismatch screen of Figure 2a and 2d and tested for genome-wide off-target events by GUIDE-seq. **a**, Genome-wide off-target data fit appropriately into the results of the mismatch screen for the targets tested. Heatmap presented here is a segment from Figure 2d. The color of the rectangular frame indicates the overall on-target specificity of an IFN on the given target as indicated in panel (c) (see calculations in Supplementary Table 6). Schematical heatmap of Figure 2a on the left shows the position of the selected EGFP target sites within the ranking. **b**, Bar chart of the total number of off-target sites detected by GUIDE-seq for data shown in panels (a). **c**, Bar chart of the overall on-target cleavage specificity expressed by the percentages of the on-target reads captured by GUIDE-seq. a-c, Data is related to Supplementary

Figure 5. Target sequences, raw, processed and heatmap disruption data and GUIDE-seq data are reported in Supplementary Tables 1-3, 6.

7. There is a mistake in Figure 5A. Supercoil plasmid should be the smaller and intense band, while the upper one is nicked plasmid.

Response: Thanks for catching this. We have corrected it (now Fig. 4).

8. Other small comments: (a) The resolution of figures is generally low, which makes it difficult to evaluate the details. (b) The figures should be better ordered. E.g., Heatmap scalebar legend for activity and specificity in Figure 2 and 4 were placed above the figure. Figure 5, the way of ordering the panel a, b, c looks very stochastic.

Response: The resolution of the figure was drastically reduced during PDF conversion. This problem have now been fixed. We also reordered the heatmap scalebar legends and panels in the figures to make them easier to follow.

(c) Color code for on-target efficiency range in Figure 7, 8, 10 is not easy. Consider using range instead, for example, 0-20%, 20-50%, 50-70%, >70%.

Response: We changed it as the reviewer suggested.

We thank Reviewer #2 for their comments, that helped to improve our manuscript.

Reviewer #3 (Remarks to the Author):

In their manuscript, Kulcsar and colleagues describe a method to identify Cas9 variants (out of a set of 19, including several newly described variants) that have a balanced sense of on- and off-target activities. In concept, this is a really nice study, and it would be great for the field to have a predictable way to tune on- and off-target activity. However, this manuscript, at least as presented, does not appear to describe such a method. While a 'cleavage rule' is motivated and an experimental algorithm is described (Figure 7a), the cleavage rule comes with many caveats (five are presented by the authors themselves in the discussion), and the algorithm is not followed in the manuscript and also comes with caveats (several presented in the discussion). Therefore, it is difficult to see the value of the authors cleavage rule.

We try to understand the target-dependence of the IFNs efficiency and specificity, the cleavage rule. Instead of enforcing a framework that does not fit the reality, we try to describe the real nature of the cleavage rule and understand the underlying reasons determining its characteristic rather than identifying caveats.

Major comments:

1) The title of this manuscript makes a broad claim - that the authors are describing a method to 'edit any target without genome-wide off-target effects.' This manuscript does not show this, and the authors need to revise the title for this manuscript to be publishable

Response: We fully agree with the reviewer that editing without off-target effects depends on the sensitivity of detection. Thus, by looking at it with another method or by screening more cells, one might find a target that can still only be edited with off-targets. Here, we would like to mention the great paper by Kleinstiver et al. entitled "High-fidelity CRISPR-Cas9 nucleases with no detectable genome-wide off-target effects", in which the authors themselves showed in the paper that the claim is in fact not completely true, not for all targets. Additionally, they tested on fewer targets than we did, with similar limitations. Nevertheless, readers understood this, and the title accurately indicated what a milestone this work represents in increasing the specificity of gene editing. With all due modesty, we believe that our work also represents a very significant step forward in the realization of off-target gene editing, although our claim is obviously also limited; however, readers will understand this in the context of ongoing scientific work, in the circumstances of a typical genome modification experiment. We cannot, of course, examine every single target and determine the target matched IFN that will provide editing without off-target. However, we demonstrated the principle that makes it reasonable to believe that, in practice, off-target-free editing can be achieved for the vast majority of targets of any cleavability rank with the set of IFNs we have developed. Furthermore, by following this path, additional missing-fidelity variants can be created if there is no optimal variant for some targets in the existing set as we found exactly this for the *CCR5* site 11. Although we demonstrated this on a relatively small number of targets, these are the targets that have not yet been successfully edited without genome-wide off-target by anyone, even though several studies have attempted to do so.

The title has been modified accordingly, and we respectfully request the acceptance of the modified title, which we have tried to refine while retaining the core message. It now reads as "Target-matched increased-fidelity SpCas9 variants potentiate the cleavage of any target without genome-wide off-target effects".

Kleinstiver, B.P., Pattanayak, V., Prew, M.S., Tsai, S.Q., Nguyen, N.T., Zheng, Z., and Joung, J.K. (2016). High-fidelity CRISPR-Cas9 nucleases with no detectable genome-wide off-target effects. *Nature* 529, 490-495.

In the authors discussion about the use of GUIDE-seq to quantify off-targets, they state that the sensitivity of GUIDE-seq is 'compatible to or even higher than that of amplicon sequencing' and cite reference 59. This is a misreading of the reference, which clearly states 'When more sensitive validation methods are used, it is clear that the complementary biochemical methods can detect some bona fide sites of off-target activity that are not picked up by GUIDE-seq. 'These types of methods are recommended when the highest sensitivity is required or in cell types that are sensitive to being transfected with dsODN tags.'

Response: We agree with the reviewer and apologize for the incorrect reference. The correct reference to support this statement is: Zischewski, Jet al., 2017.

However, we may also point out here, that GUIDE-seq and *in vitro* methods, such as Digenome-seq, CIRCLE-seq typically differ in the validated low-read count off-targets with WT SpCas9 (Tsai, S. Q. et al., 2017). The target-matched, increased-fidelity variant that ultimately succeeds in editing without off-targets reduces the cleavage of even the most persistent off-targets to zero, these are ones that typically show high-read counts with both GUIDE-seq and *in vitro* methods with the WT SpCas9. Thus, the off-targets with low-read count with WT that may be seen in *in vitro* methods but not in GUIDE-seq have low relevance for our experiments. Furthermore, the sensitivity of the biochemical methods in our case is ultimately identical to amplicon sequencing, since the off-targets found in Digenome-seq and other biochemical methods need to be validated by amplicon sequencing.

Additionally, in the revised version, we performed NGS on the top three off-target sites found by GUIDE-seq, which thus, plausibly provides the same sensitivity as the biochemical methods.

Zischewski, J., Fischer, R. & Bortesi, L. Detection of on-target and off-target mutations generated by CRISPR/Cas9 and other sequence-specific nucleases. *Biotechnol Adv* 35, 95-104, doi:10.1016/j.biotechadv.2016.12.003 (2017)

Tsai, S. Q. et al. CIRCLE-seq: a highly sensitive *in vitro* screen for genome-wide CRISPR-Cas9 nuclease off-targets. *Nat Methods* 14, 607-614, doi:10.1038/nmeth.4278 (2017).

GUIDE-seq has higher specificity for bona fide off-target loci, but it is well known that it is less sensitive than Digenome-seq, CIRCLE-seq, and CHANGE-seq, and therefore GUIDE-seq alone cannot be used to make the statement that there are no 'genome-wide off-target effects.' Frankly, even those methods paired with locus-by-locus validation cannot support the statement 'no genome-wide off-target effects', since that is a question of analytical sensitivity due to cellular input and sequencing error.

Response: We fully agree with the reviewer, as explained above in the context of the title, that strictly speaking one can never say that a target is cut without off-targets. However, the term is often used in the literature in the context of an experiment, and we have used it in such a context. Furthermore, while we agree that biochemical methods may be more sensitive to finding some bona fide off-targets, as argued in our previous response, in this particular case, the biochemical methods do not promise higher sensitivity.

2) In terms of real-world application (Figure 7), there is a disconnect between what is written in the text (lines 312-323) and what is shown in Figure 7b.

2a) In step 1, the user should screen with e-plus, B-Hypa, B-HypaR, according to the authors. This is done in Figure 7b. Following this, one is supposed to 'identify which one has the highest fidelity, that still shows sufficient efficiency.' Then one is supposed to test on-target activity of a few additional IFNs, neighboring the one selected in the first step;

For HEK site 1, all four screening nucleases had $\geq 85\%$ efficiency, so the authors presumably took the next four on the list. This should be explained in the text why they went with the next four rather than flanking four.

For HEK site 2, the choice of e-plus seems reasonable, as does the choice to take all five nucleases between e-plus and B-Hypa, though this is a lot of screening to do. There is a drop off from 105% to 25% in the first screen.

For HEK site 3, the choice of the 2nd screen does not make sense. E-plus was 92% and B-Hypa was 67%. The reasonable choice would have been to test the same five enzymes that were tested for HEK site 2. The authors need to explain this choice, since otherwise the data seem cherry-picked. For example, despite the cleavage rule, B-Hypa is at 67% and HypaR is at 95%, while the next in line (B-HypaR) is at 32%. I would not have screened HypaR, based on what is written in the text, but the authors end up choosing this as the optimal enzyme.

For VEGFA site 1, the 2nd screen choice also does not make sense. I would have chosen the five enzymes between wt and e-plus. And this is also a case where the cleavage rule has failed, since the on-target values for the five sites chosen (in order) are 69%, 98%, 126%, 86%, 78%, and these are bracketed by 73% and 51% from the first screen.

2ai) The data for HEK site 1 and VEGFA site 1 suggest that the authors may have chosen the wrong four enzymes as the first screen, or at least need to include one from the highest fidelity group (Sniper, Blackjack, HiFi, B-Sniper, B-Hifi). If the cleavage rule works well, I would expect one of those would have a higher efficiency than the 87%, 91%, and 85% (or 73%, 51%, 53%) from e-plus, B-Hypa, B-HypaR.

2b) The final step is "out of these variants, the optimal target-matched variant with the maximum specificity is selected and/or confirmed by GUIDE-seq." There is no comment in the main text about how the authors chose "optimal target-matched variant."

For HEK site 1, the authors chose B-HypaR, which had the highest rate (85%) of the 2nd screen nucleases, and it had no GUIDE-seq off-targets. This seems like a straightforward case.

For HEK site 2, the authors chose to test Hypa (91%) and B-HF1 (72%). I suppose this could be rationalized by the fact that these are the two least highly active of the 2nd screen, and both have no GUIDE-seq off-targets. This seems like a win.

For HEK site 3, as mentioned above, the authors make an unexpected choice of HypaR in the second screen. It has 95% activity, but would not have been a reasonable choice based on the cleavage rule, since it is ranked lower than the screening B-Hypa nuclease at 67%. A more consistent choice for GUIDE-seq would have been B-HF1, at 76%, and Hypa at 95%. It will be important for the authors to explain these decisions and to perform GUIDE-seq for HEK site 3 with Hypa and B-HF1.

For VEGFA site 1, the authors also made an unexpected choice of B-HypaR. First, as mentioned above, the cleavage rule has not worked, as there seems to be no trend in cleavage rate. That said, it may be reasonable to choose B-HF1, and then flank it with B-Hypa for the purposes of comparison for the paper. Both, however, have off-target sites by GUIDE-seq (see comment 3). Unexpectedly, the authors also tested B-HypaR, even though it should have been screened out after the 1st screen, and this enzyme had no GUIDE-seq off-targets. The authors need to clarify these decisions, since they contradict what would be expected by the method described in the manuscript. Additional GUIDE-seq data may be useful as well.

Response: We sincerely thank the reviewer for his thorough and careful critique, with which we can agree with in many respects, although a significant part of the criticism stems from a misunderstanding of the cleavage rule and a lack of adequate explanation of our method on our part. We have tried to make it more understandable in the revised manuscript (page 16).

We have also refined the algorithm itself. Now, (i) we use B-HF1 in the first screen, which makes the number of IFNs within the regions slightly more proportional for the second screen, and (ii) as there are several IFNs in the low-fidelity IFN region with minimal fidelity differences, we omitted two low-fidelity IFNs (Sniper and HiFi). This reduced the number of IFNs to screen. For the application of the method, we clarified the following: in the second screen, we examined variants upstream of the IFN with the highest fidelity that still worked (not the flanking ones) in the hope of finding a working variant with an even higher fidelity. This type of second screen choice proved advantageous in most cases, except for the AR target (in Figure 7), where the failure to find a higher fidelity IFN left us with only one variant after the second screen for the GUIDE-seq experiment, but it did meet our expectations and was edited without off-target. With GUIDE-seq we tested the two highest ranking IFNs that still worked (>50%). We included these considerations in the figure legend of the revised version of Figure 6 in agreement with the reviewer's advice.

However, we also try to clarify here, that there does not necessarily need to be a monotonically decreasing trend between working nucleases (>50%) based on the cleavage rule, as the cleavage rule basically distinguishes whether an IFN cleaves or does not cleave a target. We discuss this in more detail on page 19, lines 434 of the manuscript and in our response to the first reviewer's comment number 3. We also briefly summarize it here: The most obvious interpretation of the emergence of the cleavage rule is that the three main factors, the contribution of the target sequence, the effect of fidelity-increasing mutations and the effect of mismatches (for off-targets), add up to produce the pattern characteristic of the cleavage rule. To achieve this, all three are expected to act at the same step of Cas9s function, with similar mechanisms.

For mismatches and fidelity increasing mutations in eSpCas9 and SpCas9-HF1, we know the mechanism by which they act: they keep the HNH domain in an inactive conformation and slow its docking into the cleavage competent conformation (Chen et al., 2017; Dagdas et al., 2017; Yang et al, 2018). This implies that the contribution of target sequences also affects the docking of the HNH domain. It is known that the formation of a spacer-target DNA hybrid helix activates the nuclease by inducing docking of the HNH domain into the cleavage-competent conformation (Chen et al., 2017; Dagdas et al., 2017; Yang et al, 2018). Our results suggest that the extent of the activation may differ depending on the sequence of the target.

This mechanism is of interest here, because it explains why the appearance of the cleavage rule in the cleavage pattern of a target is not always apparent amongst active nucleases. When an IFN efficiently cleaves a target sequence, the HNH domain is stably docked in the cleavage-competent conformation. Thus, the other active IFNs cannot substantially increase the cleavage of that target, because once the HNH domain is stably docked in the

cleavage-competent conformation, it is not possible to significantly increase its docking further. Thus, the trend expected from the cleavage rule is not always observed in the cleavage % of working IFNs, and in its absence, the effect of other factors that usually only slightly alter the cleavage efficiency of the variant becomes more pronounced. This is a reasonable explanation for the fact, also noted by the reviewer, that the cleavage rule is often not apparent in the cleavage percentage of working IFNs on a target. We do not see this as a caveat, but rather as the true nature of the cleavage rule.

In these experiments, for practical reasons, we set the threshold between working and non-working conditions (for a variant normalized activity on a given target) at 50% (rather than at the more relevant 20% used when establishing the cleavage rule) to make the variants in the working category as usable as possible in real-life applications. This may give the impression that the cleavage rule does not work for two targets (HEK site 2 and VEGFA site 3, with their cleavage being just below 50%). Although there are occasional exceptions elsewhere, these tend to reflect the high threshold selection.

Figure 6

Figure 6. The optimal, target-matched SpCas9 nuclease, which shows efficient on-target editing and no off-target effects, is identified for each target using a two-step approach

a, Schematic representation of the two-step screening method used on a hypothetical target example. The first panel shows the on- and off-target activity of a set of IFNs with increasing fidelity on a hypothetical target example. The second panel shows the screening method, which identifies the optimal variant for the target without having to test all of the variants. In the first step, a rough on-target screen is performed, where the WT and three selected IFNs, that divide the target ranking range into four approximately proportional sections, are tested. The second step is a fine-tuning on-target screen, that involves the not yet used variants with higher fidelity than the highest ranking active (green) variant from the first screen, and it identifies the target-matched variants (active variants with the highest fidelity). If necessary, two sufficiently active (here their normalized activity is above 50%, but this may depend on the application under consideration) target-matched variants can be screened for the absence of genome-wide off-targets. b, The identification of the target-matched variants that provide appropriate editing without any genome-wide off-target is demonstrated on three targets that had been tested in Tsai et al. The numbers in the colored Cas protein illustrations indicate the percentage value of the on-target genome modifications normalized to WT (measured by NGS). Colored circles indicate whether a target was edited with (red) or without (green) off-targets in the GUIDE-seq experiment. The total number of off-target sites detected by GUIDE-seq are shown for each target in bar charts on the right side of the panel. Data related to Figure 8, Supplementary Figures 10 and 11. Target sequences, NGS and GUIDE-seq data are reported in Supplementary Tables 1, 5 and 6.

Figure 7

Figure 7. Even repetitive, non-typical sequences can be edited without off-targets by employing optimal target-matched IFNs

Targets shown here are a collection of targets that had previously only been edited by IFNs with off-targets detected by GUIDE-seq. Here, they were all successfully edited without any genome-wide off-targets when assessed by GUIDE-seq using target-matched IFNs. The numbers in the colored Cas protein illustrations indicate the percentage value of the on-target genome modifications normalized to WT (measured by NGS). GUIDE-seq was performed with one or two target-matched IFNs that reached at least 50% normalized on-target editing. Colored circles indicate whether a target was edited with (red) or without (green) off-targets in the GUIDE-seq experiment. Some targets can be edited with no detectable genome-wide off-targets by more than one target-matched IFN, or in other cases by an IFN in RNP form that can further increase specificity. Bar charts of the total number of off-target sites detected by GUIDE-seq are shown on the right side of the panel. Data related to Figure 8, Supplementary Figures 10 and 11. Target sequences, NGS and GUIDE-seq data are reported in Supplementary Tables 1, 5 and 6.

Chen, J. S. et al. Enhanced proofreading governs CRISPR-Cas9 targeting accuracy. Nature 550, 407-410, doi:10.1038/nature24268 (2017).

Dagdaz, Y. S., Chen, J. S., Sternberg, S. H., Doudna, J. A. & Yildiz, A. A conformational checkpoint between DNA binding and cleavage by CRISPR-Cas9. *Sci Adv* 3, eaao0027, doi:10.1126/sciadv.aao0027 (2017).

Yang, M. et al. The Conformational Dynamics of Cas9 Governing DNA Cleavage Are Revealed by Single-Molecule FRET. *Cell Rep* 22, 372-382, doi:10.1016/j.celrep.2017.12.048 (2018).

3) The authors claim that 'Figure 7b demonstrates that with the panel of 19 IFNs, all four targets could be edited without any genome-wide off-target effect detected by GUIDE-seq.' This is a well-parsed, factually-correct statement. But it is not what the authors describe in Figure 7a. It is not reasonable to expect a lab to do GUIDE-seq with 19 different Cas9 variants and then choose the best one. Or, if it is, labs can already do this with a large subset of the 19, without relying on this manuscript. Even if one does the full method, as described, it would involve testing 8-9 variants, and doing GUIDE-seq on 2-3 of them.

Response: This is one of the messages of the manuscript; unfortunately, there is no silver bullet due to the magnitude of the differences in sequence contributions relative to the effects of mismatches as described: variants with different fidelities are appropriate for different targets to achieve maximum specificity. Using the method described in the manuscript, a relatively large increase can be achieved in specificity using only the first screen, i.e. testing 3 IFNs. But based on our work, we already know which three are worth using, what to expect from this screen, and based on that, we know how to proceed if we want even higher specificity. We also slightly modified the method itself in the revised manuscript (revised Fig. 6a) to make it more feasible with a little less IFN testing.

Based on the recognition of the contribution of the target sequences, we have designed new IFNs in this manuscript, the benefit of which is also apparent. For example, in Figure 7, a newly designed IFN achieved maximum specificity (no genome-wide off-target by GUIDE-seq) in 5 cases, compared to 2 cases when an IFN (evoSpCas9) from the conventional set did so.

We agree with the reviewer that there are many alternative algorithms and possible IFN subsets to perform the first and second screens to find a target-matched IFN for a given target. They also require testing a similar number of IFNs and the knowledge and IFNs presented in this manuscript. For the set now modified to 17, testing 5-7 IFNs using amplicon sequencing will generally be sufficient to identify a target-matched IFN.

3a) The experiments in Figure 8 do not seem to follow at all the strategy described in Figure 7a, so it is hard to assess whether the authors method works. A comment is made in the figure legend that three targets 'had sufficient information to skip the rough screen'. A similar comment is made in the main text. The authors don't explicitly state what that information is, but in any case, since the authors are essentially presenting the method described in Figure 7a, that exact method should be used in Figure 8.

Response: These targets are those for which off-target editing has already been attempted with several IFNs, such as SpCas9-HF1, Hypa or evoSpCas9, some of which have even been tested with additional IFNs. IFNs that have been applied to a given target in previous work are indicated in the original Figure 9 (revised Fig. 8; or the exact results for these targets are summarized in Supplementary Table 6: Data from other studies). If it was already known from the literature, that a target did not work with evo, for example, but did work with Hypa, we only tried to test the intermediate IFNs. Our goal was not to demonstrate the method, but to find one IFN that fit the targets.

It seems that this may have created some level of confusion, so we have omitted this information from the revised Figure 8 and, as the reviewer requested, we also performed the first screen on these targets ourselves

(revised Fig. 7, former Fig. 8). We trust that the reviewer is right and that our message is now easier to understand.

Figure 8

Figure 8. Target-matched nucleases show high efficiency without any genome-wide off-target for targets tested in this study regardless their ranking

Summary of targets edited by IFNs, examined in this study with GUIDE-seq and NGS. For 10 target sites, including those challenging targets where previous attempts with IFNs had failed, we were able to perform editing without any off-target detected by GUIDE-seq and further confirmed by NGS on the top three site. For the highest ranked target, CCR5 site 11, NGS still identified residual off-target activity even with the highest ranked B-HeF, indicating that development of an even higher fidelity IFN would be beneficial for accessing the highest cleavability rank. The colors of the squares indicate the percentage value of the on-target genome modification normalized to WT (measured by NGS). Colored circles indicate the summarized GUIDE-seq and NGS results; green circle indicates when both NGS and GUIDE-seq showed no off-targets, red circle indicates off-target editing detected either by GUIDE-seq or NGS, light green circle indicates when no off-target was found but it was only tested by NGS and gray circle indicate no data. Off-target editing data of these targets (GUIDE-seq experiments) from the literature are summarized in Supplementary Table 6: Data from other studies. The ranking of the targets is weakly related to either b, the number of their predicted off-target sites, or c, the detected off-target sites using WT SpCas9 (for details see Supplementary Table 10). Data are related to Figures 6, 7 and 9, Supplementary Figures 9, 10 and 11.

3b) The data in Figure 8 for CCR5 site 11, FANCF site 2, and VEGFA site 3 also do not appear to follow the cleavage rule. I would expect there to be a lot more correlation than there is between adjacent nucleases (see comment 4). I also would expect that more of the targets at the border between green and blue or green and orange would have an absence of GUIDE-seq off-targets, if the cleavage rule is relevant, and this does not seem to be the case.

Response: These are the targets for which we used RNP, and therefore are difficult to fit into the plasmid-based ranking order.

4) Figure 9 actually cleanly shows that the authors did not follow their recommendations of using WT, e-plus, B-Hypa, and B-HypaR as the first screening set.

Response: The reviewer is right, we used the knowledge from earlier studies, but did not follow the recommendations. As we pointed out in a previous comment, most of these targets were chosen for testing because no IFN has managed to edit them efficiently and without off-target effects. The old

Figure 9 shows which IFNs had been tested in previous studies and failed for each target. Knowing the cleavage rule, this information allows us to skip the first screen and find the target-matched variants by testing fewer IFNs. So this actually demonstrated the benefit of knowing the cleavage rule. Nevertheless, in the revised version, we did a first screen to demonstrate our approach on more examples and changed the old Figure 9 (now Figure 8).

5) The authors seem to have done a very limited set of off-target sites to establish the cleavage rule (looking at mismatches in only the first few positions in the target site). To establish that it is going to be generalizable, the authors should extend this analysis through all positions of the target site. This may contribute to some of the cleavage rules failures that are seen in Figures 7 and 8.

Response: As we tried to clarify in the previous comments, the reviewer's observation does not really mean that the cleavage rule fails. We examined PAM-distal mismatches at positions 14-19 that have the least impact on IFN activity and are therefore the most sensitive indicators of mismatch effects. Furthermore, analyses of the Kim off-target data included all possible mismatch positions, supporting the idea that the cleavage rule can be generalized in this regard as well. While we also recognize that mismatches from the distal and proximal PAM positions may have different effects on SpCas9 activity and agree with the reviewer that more data will allow for more robust conclusions, we do not see here that examining another 300 mismatched guides beyond the 1,800 analyzed and 162 examined would provide results proportional to the effort required.

6) I agree that off-target activity is a concern, but I think language like 'raises potential safety concerns' is more accurate than 'poses safety concerns' (lines 40-41), since the functional consequences of off-target editing are yet to be understood. Similarly, the phrase 'pose a major limitation' (line 17) may be overstated, as clearly clinical trials are being allowed to move forward, and not all are using a high-fidelity Cas9.

Response: We changed the wording, as the reviewer suggested.

We thank Reviewer #3 for their comments, that helped to improve our manuscript.

Reviewers' Comments:

Reviewer #1:

Remarks to the Author:

My comments were adequately addressed and the authors have increased the quality of the manuscript with their edits and additional data.

Reviewer #2:

Remarks to the Author:

Peter et al., has addressed my comments and concerns in their revised manuscript. I enjoy reading this nice manuscript. Their findings provide an experimental approach to select the best possible Cas9 variant with high editing efficiency and specificity. A few minor changes are still remained to be corrected carefully.

1. Data availability. Although the raw and process read counts have been provided in the supplementary dataset, the raw NGS sequencing data (fastq) for the targeted amplicon sequencing and GUIDE-seq should be provided with accession code to a public data depository such as GEO. These are important data to be shared within the scientific community.
2. Figure S1 and in the methods section, the term of "viable/live" used to gating the cells. Unless PI or other type of staining is used for detecting live cells, this word should be removed. SSA and FSC are not enough to distinguish live and dead cells.
3. Figure S5. The off-target data for EGFPs43 seems to be exceptionally high. The on-target reads are far lower than several off-targets, which contain more than 4 mismatches. G-A wobble mismatch could explain part of it, but not fully. All four gRNAs are EGFP targeting ones. What are the expected number of potential off-target site in the genome, if allowing up to 5-6 mismatches?
4. Figure S6. Texts in the images are not readable.
5. Figure S7. Data from the WT Cas9 for some EGFP sites have been replotted in sub figure a, b, c, d, and e. These should be mentioned in the figure legend.
6. Figure S8. What are the nontargeted 1 and nontargeted 2?
7. Methods. Texts in line 549-555 should be in the results section.

Most of the comments from reviewer 3 have been properly addressed. A few comments are remained to be further addressed.

A. Regarding to the point of no genome-wide off-target effects. It is important to put this in context. I agree with the reviewer 3 that the authors should carefully revise the conclusion drawn on "no" genome-wide off-targets. It is really related to the methods used for detection. The author in their response letter "point 3" has already acknowledged that "one can never say that a target is cut without off-targets". This is really related to the off-target detection methods used, the experiment setups, ways of delivery. I think that there is no need for the authors to continue the overstatements from the previous publications. For the cited articles, these are from the very beginning of the CRISPR era, when genome-wide off-target evaluation methods are still limited. An alternative is to use the word "detectable" to limit the statement to context. The word of "any target" should be removed. Here are a few suggestions for the titles:

Long version

1. Target-matched increased-fidelity SpCas9 variants potentiate gene editing with high efficiency and no detectable genome-wide off-target effects.
2. A cleavage rule for selection of increased-fidelity SpCas9 variants with high on-target efficiency

and no detectable genome-wide off-target effects.

Short version

3. Target-matched increased-fidelity SpCas9 variants potentiate gene editing with high efficiency and specificity.

4. A cleavage rule for selection of increased-fidelity SpCas9 variants with high efficiency and specificity.

The cleavage rule is a key finding of the study. It is worthy to highlight that in the title.

B. Agreed with reviewer's concern in point 3b, regarding the RNP delivery that will not fit into the plasmid-based ranking order, if this is true, this will greatly limit the application of the current cleavage rule found by the study? What if the Cas9 variant delivered by lentivirus, AAV, and other form. This was not reflected, rather in contrast in the discussion section. Line 487-488, "target-matched IFNs are compatible with many other approaches (e.g. RNP form or dRNA)". If correct, one would need generate an order of cleavage rule for each delivery form.

C. After printing out the manuscript, the color code for the on-target efficiency range 100-75% and 75-50% is extremely difficult to be distinguished. This can be visually misleading regarding the on-target efficiency, as 50% will look similar to 100%. These are presented in Figure 6-9.

Reviewer #1 (Remarks to the Author):

My comments were adequately addressed and the authors have increased the quality of the manuscript with their edits and additional data.

Thanks for your positive feedback!

Reviewer #2 (Remarks to the Author):

Peter et al., has addressed my comments and concerns in their revised manuscript. I enjoy reading this nice manuscript. Their findings provide an experimental approach to select the best possible Cas9 variant with high editing efficiency and specificity. A few minor changes are still remained to be corrected carefully.

1. Data availability. Although the raw and process read counts have been provided in the supplementary dataset, the raw NGS sequencing data (fastq) for the targeted amplicon sequencing and GUIDE-seq should be provided with accession code to a public data depository such as GEO. These are important data to be shared within the scientific community.

We have uploaded the NGS data to SRA, under the submission number: SUB13704883, we submitted all relevant plasmids to Addgene. Their numbers are indicated in the manuscript (Methods section and Supplementary Table 1).

2. Figure S1 and in the methods section, the term of “viable/live” used to gating the cells. Unless PI or other type of staining is used for detecting live cells, this word should be removed. SSA and FSC are not enough to distinguish live and dead cells.

We agree with the reviewer on this point, corrected it in the manuscript.

3. Figure S5. The off-target data for EGFPs43 seems to be exceptionally high. The on-target reads are far lower than several off-targets, which contain more than 4 mismatches. G-A wobble mismatch could explain part of it, but not fully. All four gRNAs are EGFP targeting ones. What are the expected number of potential off-target site in the genome, if allowing up to 5-6 mismatches?

Neither the ranking of the targets nor their GUIDE-seq detected off-target sites for WT SpCas9 is related to the number of their *in silico* predicted off-target sites (see table below and Supplementary Figure 5b). Same conclusion as in Figure 8 for genomic target sites.

site	mismatches to on-target site						total
	1	2	3	4	5	6	
EGFP site 10	0	0	2	78	875	6790	7745
EGFP site 11	0	0	6	69	1021	8733	9829
EGFP site 15	0	3	10	158	1377	9833	11381
EGFP site 43	0	0	5	88	827	5895	6815

Data in table from Supplementary Table 10.

4. Figure S6. Texts in the images are not readable.

Corrected

5. Figure S7. Data from the WT Cas9 for some EGFP sites have been replotted in sub figure a, b, c, d, and e. These should be mentioned in the figure legend.

In the revised manuscript, this is now indicated in the figure legend.

6. Figure S8. What are the nontargeted 1 and nontargeted 2?

Thanks for catching it! We corrected the figure.

7. Methods. Texts in line 549-555 should be in the results section.

Thanks for the suggestion, we included it in the Discussion section. In our opinion it fits better there.

Most of the comments from reviewer 3 have been properly addressed. A few comments are remained to be further addressed.

A. Regarding to the point of no genome-wide off-target effects. It is important to put this in context. I agree with the reviewer 3 that the authors should carefully revise the conclusion drawn on “no” genome-wide off-targets. It is really related to the methods used for detection. The author in their response letter “point 3” has already acknowledged that “one can never say that a target is cut without off-targets”. This is really related to the off-target detection methods used, the experiment setups, ways of delivery. I think that there is no need for the authors to continue the overstatements from the previous publications. For the cited articles, these are from the very beginning of the CRISPR era, when genome-wide off-target evaluation methods are still limited. An alternative is to use the word “detectable” to limit the statement to context. The word of “any target” should be removed. Here are a few suggestions for the titles:

Long version

1. Target-matched increased-fidelity SpCas9 variants potentiate gene editing with high efficiency and no detectable genome-wide off-target effects.
2. A cleavage rule for selection of increased-fidelity SpCas9 variants with high on-target efficiency and no detectable genome-wide off-target effects.

Short version

3. Target-matched increased-fidelity SpCas9 variants potentiate gene editing with high efficiency and specificity.
4. A cleavage rule for selection of increased-fidelity SpCas9 variants with high efficiency and specificity.

The cleavage rule is a key finding of the study. It is worthy to highlight that in the title.

Thank you for your suggestions. To comply with the Nature Communications policy, we have shortened one of the proposed title to the following: "A cleavage rule for selection of increased-fidelity SpCas9 variants with high efficiency and no detectable off-targets".

B. Agreed with reviewer's concern in point 3b, regarding the RNP delivery that will not fit into the plasmid-based ranking order, if this is true, this will greatly limit the application of the current cleavage rule found by the study? What if the Cas9 variant delivered by lentivirus, AAV, and other form. This was not reflected, rather in contrast in the discussion section. Line 487-488, "target-matched IFNs are compatible with many other approaches (e.g. RNP form or dRNA)". If correct, one would need generate an order of cleavage rule for each delivery form.

Thank you for addressing this point; we are striving to enhance clarity in the revised manuscript. Indeed, the RNP data cannot be simply integrated into the plasmid delivery format. As demonstrated in Supplementary Figures 8a and b, the order of both the targets and IFNs remains unchanged during RNP delivery. However, the nuclease that best matches a specific target (i.e., the optimal nuclease for a given target) may vary when the delivery method changes. This observation aligns with the fact that RNP delivery leads to lower nuclease levels in cells for a shorter duration, resulting in reduced detectable cellular IFN activity. Consequently, the activity threshold that distinguishes effective and non-effective conditions for a given IFN may shift upwards across the target sequences/ranking.

Moreover, due to the aforementioned reasons, it is plausible that for a given target nucleases ranked lower in fidelity might already achieve editing without detectable off-targets. Thus, the determination of the target-matching nuclease should naturally be in the input format intended for use (i.e if virus delivery is needed, it should be identified with the set of variants in virus delivery). For maximum specificity, once the target-matched nuclease has been identified through plasmid delivery, it may be evaluated along with one or two lower fidelity neighbors in the IFN ranking using RNP form or dRNA approach.

We have incorporated several sentences to clarify these points at lines 494-500.

C. After printing out the manuscript, the color code for the on-target efficiency range 100-75% and 75-50% is extremely difficult to be distinguished. This can be visually misleading regarding the on-target efficiency, as 50% will look similar to 100%. These are presented in Figure 6-9.

Thank you for the input. We changed the color code for the on-target efficiency range 75-50% and now it can be distinguished when printed out in Figure 6-9 and Supplementary Figure 10.

We thank reviewer #2 for his thorough comments, which helped us improve our manuscript, and for taking the extra effort to review our responds to the comments of reviewer #3!